# Fe/Mg-Modified Carbonate Apatite with Uniform Particle Size and Unique Transport Protein-Related Protein Corona Efficiently Delivers Doxorubicin into Breast Cancer Cells

**DOI:** 10.3390/nano10050834

**Published:** 2020-04-27

**Authors:** Sheikh Tanzina Haque, Md. Emranul Karim, Syafiq Asnawi Zainal Abidin, Iekhsan Othman, Mark M. Banaszak Holl, Ezharul Hoque Chowdhury

**Affiliations:** 1Jeffrey Cheah School of Medicine and Health Sciences, Monash University Malaysia, Jalan Lagoon Selatan, Bandar Sunway, Subang Jaya 47500, Selangor, Malaysia; Sheikh.Haque@monash.edu (S.T.H.); karim604306@gmail.com (M.E.K.); syafiqnawi@gmail.com (S.A.Z.A.); Iekhsan.Othman@monash.edu (I.O.); 2Chemical Engineering, Monash University, Clayton 3800, VIC, Australia; mark.banaszakholl@monash.edu; 3Health and Wellbeing cluster, Global Asia in the 21st Century (GA21) Platform, Jeffrey Cheah School of Medicine and Health Sciences, Monash University Malaysia, Jalan Lagoon Selatan, Bandar Sunway, Subang Jaya 47500, Selangor, Malaysia

**Keywords:** nanoparticles (NPs), inorganic NPs, doxorubicin (DOX), breast cancer, carbonate apatite (CA), Fe/Mg-carbonate apatite (Fe/Mg-CA), pH responsive drug delivery, cytotoxicity, cellular uptake, protein corona, divalent and trivalent cations, biodistribution, blood plasma

## Abstract

Breast cancer is the abnormal, uncontrollable proliferation of cells in the breast. Conventional treatment modalities like chemotherapy induce deteriorating side effects on healthy cells. Non-viral inorganic nanoparticles (NPs) confer exclusive characteristics, such as, stability, controllable shape and size, facile surface modification, and unique magnetic and optical properties which make them attractive drug carriers. Among them, carbonate apatite (CA) particles are pH-responsive in nature, enabling rapid intracellular drug release, but are typically heterogeneous with the tendency to self-aggregate. Here, we modified the nano-carrier by partially substituting Ca^2+^ with Mg^2+^ and Fe^3+^ into a basic lattice structure of CA, forming Fe/Mg-carbonate apatite (Fe/Mg-CA) NPs with the ability to mitigate self-aggregation, form unique protein corona in the presence of serum and efficiently deliver doxorubicin (DOX), an anti-cancer drug into breast cancer cells. Two formulations of Fe/Mg-CA NPs were generated by adding different concentrations of Fe^3+^ and Mg^2+^ along with a fixed amount of Ca^2+^ in bicarbonate buffered DMEM (Dulbecco’s Modified Eagle’s Medium), followed by 30 min incubation at 37 °C. Particles were characterized by turbidity analysis, z-average diameter and zeta potential measurement, optical microscopy, field emission scanning electron microscopy (FESEM), Fourier transform infrared spectroscopy (FTIR), energy dispersive X-ray (EDX), flame atomic absorption spectroscopy (FAAS), pH dissolution, drug binding, cellular uptake, thiazolyl blue tetrazolium bromide (MTT) assay, stability analysis, and protein corona study by LCMS (Liquid chromatography-mass spectrometry). Both formulations of Fe/Mg-CA displayed mostly uniform nano-sized particles with less tendency to aggregate. The EDX and FAAS elemental analysis confirmed the weight (%) of Ca, Fe and Mg, along with their Ca/P ratio in the particles. A constant drug binding efficiency was noticed with 5 μM to 10 μM of initial DOX concentration. A pH dissolution study of Fe/Mg-CA NPs revealed the quick release of DOX in acidic pH. Enhancement of cytotoxicity for the chemotherapy drug was greater for Fe/Mg-CA NPs as compared to CA NPs, which could be explained by an increase in cellular internalization as a result of the small z-average diameter of the former. The protein corona study by LCMS demonstrated that Fe/Mg-CA NPs exhibited the highest affinity towards transport proteins without binding with opsonins. Biodistribution study was performed to study the effect of DOX-loaded Fe/Mg-CA NPs on the tissue distribution of DOX in Balb/c 4T1 tumor-bearing mice. Both formulations of Fe/Mg-CA NPs have significantly increased the accumulation of DOX in tumors. Interestingly, high Fe/Mg-CA NPs exhibited less off-target distribution compared to low Fe/Mg-CA NPs. Furthermore, the blood plasma analysis revealed prolonged blood circulation half-life of DOX-loaded low and high Fe/Mg-CA NPs compared to free DOX solution. Modifying CA NPs with Fe^3+^ and Mg^2+^, thereby, led to the generation of nano-sized particles with less tendency to aggregate, enhancing the drug binding efficiency, cellular uptake, and cytotoxicity without hampering drug release in acidic pH, while improving the circulation half-life and tumor accumulation of DOX. Therefore, Fe/Mg-CA which predominantly forms a transport protein-related protein corona could be a proficient carrier for therapeutic delivery in breast cancer.

## 1. Introduction

Breast cancer is one of the major lethal diseases with an occurrence rate of 1 in 35 [1]. It is prevalent in women occurring in the inner lining of milk-producing ducts (invasive ductal carcinoma) or the lobules (invasive lobular carcinoma) that supply the milk and spread through the lymph, bloodstream or by local invasion into the nearby tissues. Current treatment modalities, such as chemotherapy, radiotherapy and hormonal therapy produce adverse effects on healthy tissues, obtruding the treatment time and reducing the dose of therapeutics [2]. Usually, poor water solubility, limited bioavailability, non-specific biodistribution throughout the body, poor cellular internalization and inadequate drug release are attributed to low drug availability in the cytosol of target cancer cells [3]. Therefore, large drug doses are often required to maximize the therapeutic efficacy with concomitant detrimental side effects.

The development of a carrier that can successfully deliver drugs to the site of action is of paramount importance. Among the systems that showed prospective outcomes, nanoparticles (NPs), nanowires and nanotube-based systems (carbon nanotubes) are gaining interest due to their high potential to serve as effective diagnostic and therapeutic agents [4,5,6,7]. NPs used in drug delivery are usually in the range of 100–1000 nm [8]. In cancer therapy, they can mitigate adverse side effects altering the stability, absorption, and release profile of drugs. Moreover, they can enhance the site targeted delivery and cellular uptake in tumors through receptor-mediated endocytosis that may enable them to escape P-glycoprotein recognition [9] and overcome MDR (multidrug resistance), and reduce the off-target effects and systemic clearance of drugs. Specific drug delivery to cancer cells can be promoted via either active targeting in which NPs functionalized with ligands finally adhere to the receptors of target cells, or passive targeting whereby NPs pass through the leaky blood vessels of cancer cells [8].

Interestingly, the fate of the NPs in the biological environment largely depends on the “protein corona” profile of the respective NPs. After NPs are administered to a biological environment, they are immediately coated with proteins through hydrophobic or ionic interactions, forming a “protein corona” on their surface [10]. The adsorption of the proteins on the NP surface modifies the surface properties, particle size, and stability in a manner that generally defines the “biological identity” and determines the physiological responses, such as biodistribution, tissue accumulation, cellular uptake, intracellular trafficking as well as the therapeutic outcome [11].

The protein corona also performs a vital role in the identification and removal of NPs by macrophages of the reticuloendothelial system (RES) [12]. Opsonins (e.g., fibrinogen, complement factors, and IgG), once bound to NPs, stimulate phagocytosis and thus the removal of the NPs from systemic circulation [13] and sequestration in the RES organs, e.g., liver and spleen [12]. It is also possible for the protein corona to play a positive role. For example, dysopsonins such as albumin heighten circulation times in blood and can also serve to protect drug cargo from degradation [14,15]. Therefore, engineered NPs with the ability to specifically bind certain proteins can lead the NPs to a specific pathway or part of the body [13,16].

Since viral particles often exert adverse effects in terms of carcinogenicity and immunogenicity [17,18,19], non-viral vectors have been substantially explored for clinical studies. Among the non-viral vectors which can be either organic (e.g., cationic liposomes or polymers) or inorganic (e.g., carbonate apatite (CA), quantum dots, mesoporous silica, iron oxide) [20], liposomes have been investigated in many pre-clinical and clinical trials [21,22] with some common challenges such as releasing drug before reaching the tumor and instability in the physiological environment. Inorganic NPs have great promise as carriers for drugs as well as imaging agents as a result of their tunable characteristics such as high surface area to volume ratio, scalability in synthesis, controllable shape and size, facile surface modification, inertness, stability, and unique optical and magnetic properties [23]. However, inorganic NPs show limitations including non-biodegradability, leakage of drugs before reaching the target, insufficient tumor accumulation, poor cellular internalization and late endosomal release, which in concert give undesired toxicity and poor biocompatibility.

A particularly promising approach to overcoming the above challenges is the emergence of a pH-responsive drug delivery system that triggers the release of drugs inside the tumor cells or at the tumor microenvironment [24,25]. pH-responsive delivery is an effective approach for the release of anticancer drugs with a change in pH which is different from the blood serum (pH 7.4) to the extracellular tumor tissue (pH 6.8) and the intracellular endosomes (pH 4.5–5.5) and lysosomes (pH 5–6.5). The acidic pH in the tumor microenvironment is related to the elevated anaerobic metabolism needed for the growth of tumor under hypoxia in weekly perfused areas of tumor [26].

pH-sensitive CA with a basic structure consisting of Ca^2+^, PO_4_^3−^ and CO_3_^2−^ is a biodegradable inorganic non-viral vector with heterogeneous charge distribution and alikeness to the hard tissue component of the body. CA has many useful properties including nanoscale particle morphology allowing for efficient endocytosis, rapid dissolution in the acidic pH of endosomes or lysosomes, and concomitant release of therapeutics from the endosomal/lysosomal compartment into the cytoplasm [27,28,29,30,31,32]. Ca^2+^ ions are involved in cell apoptosis, survival and migration by modulating various cell signaling pathways [33]. In addition, Ca^2+^ homeostasis and signaling play a vital role in tumorigenesis. It can also promote aggregation of NPs in a dose-dependent manner. The CA NPs could be modified with other divalent or trivalent cations, (for example- Fe^3+^, Sr^2+^, Al^3+^, and Mg^2+^) to further tune their properties [34].

Among the different divalent and trivalent cations, Mg^2+^ and Fe^3+^ play quite interesting roles. Mg^2+^ is an alkaline earth metal ion which is above Ca^2+^ in the periodic table. Hence, it is less reactive than Ca^2+^. It can hinder the growth of the NPs by distorting the lattice structure of the apatite NPs by partially replacing the Ca^2+^ ions, thus, generating small crystals in a dose-dependent manner [35]. It is a vital mineral required to carry out important functions in our body [36]. On the other hand, Fe^3+^ is an important transition metal ion and because of its half-filled orbital (3d5), it is more stable than Fe^2+^. Fe^3+^ is present in the body and its main function is to form hemoglobin. Different iron (heme) binding proteins have a strong affinity towards Fe^3+^ ions, enabling the tuning of the protein corona configuration on NPs [37,38]. This effect is particularly pronounced since Fe^3+^ apparently substitutes the Ca^2+^ ions in the bulk CA structure [34]. Fe^3+^ can increase the growth of the NPs and thus further lead to self-aggregation between the particles. Since Mg^2+^ and Fe^3+^ apparently have opposing roles (i.e., Mg^2+^ inhibits the growth of the NPs, while Fe^3+^ induces the growth of the NPs), the combined effect of these ions might be beneficial in controlling the size of CA particles.

Among various anticancer drugs, hydrophilic drugs such as doxorubicin (DOX) are often preferred over hydrophobic drugs which exhibit limitations including limited bioavailability, poor solubility, and uncontrollable precipitation after dosing [39]. DOX is frequently used for treating solid tumors and hematopoietic malignancies. It is an anthracycline drug that functions by (1) intercalating DNA, thus preventing uncoiling and disrupting the DNA repair mechanism of topoisomerase enzyme and hindering DNA replication, and (2) releasing free radicals that destroy the cell membrane, DNA and proteins, leading to cell death by apoptosis. However, as DOX fosters the production of free radicals, it leads to severe side effects including congestive heart failure, nausea, and vomiting [40,41]. The drug also suffers from low tumor selectivity [42] and drug resistance [43]. To increase the therapeutic index via a combination of improved cancer cell targeting and reduced off-target effects, CA-based nano-carriers were employed to load and deliver DOX, a hydrophilic drug [44,45] to cancer cells. DOX could bind to the CA NPs through electrostatic interaction where the positive charge of DOX is likely to bind with the negative charge of CO_3_^2−^ and PO_4_^3−^ domains in apatite NPs.

In general, CA NPs produced to date have a tendency to self-aggregate, which leads to polydisperse and heterogeneous particles. Herein, we report on the development of pH-responsive Fe/Mg-carbonate apatite (Fe/Mg-CA) NPs that formed unique transport protein-related corona and enhanced the cellular uptake of DOX with the consequence of much higher cytotoxicity compared to CA NPs in breast cancer cells. Furthermore, the in vivo blood plasma analysis and biodistribution study demonstrated an improved circulation time of DOX in the blood and reduced off-target distribution. Therefore, Fe/Mg-CA which forms a transport protein-related protein corona could be a promising tool for efficient therapeutic delivery in breast cancer patients.

## 2. Materials and Methods

### 2.1. Materials

Calcium chloride dihydrate (CaCl_2_·2H_2_O), magnesium chloride hexahydrate (MgCl_2_·6H_2_O), ferric chloride hexahydrate (FeCl_3_·6H_2_O), sodium bicarbonate (NaHCO_3_), doxorubicin hydrochloride (DOX), dimethyl sulphoxide (DMSO), Dulbecco’s modified eagle medium (DMEM), thiazolyl blue tetrazolium bromide (MTT), and ethylene diamine tetraacetic acid (EDTA) were acquired from Sigma-Aldrich (St. Louis, MO, USA). Michigan Cancer Foundation-7 (MCF-7) cells were purchased from ATCC (Manassas, Virginia, USA). Trypsin-ethylene diamine tetraacetate (trypsin-EDTA), DMEM powder, penicillin-streptomycin and fetal bovine serum (FBS) were procured from Gibco by Life Technology (Thermo Fischer Scientific, Waltham, MA, USA). Hydrochloric acid (HCl) and ethanol were obtained from Fischer Scientific (Loughborough, Leicestershire, UK).

### 2.2. Fabrication of Fe/Mg-CA NPs

The bicarbonated DMEM was prepared by dissolving 0.27 g DMEM powder in 20 mL Milli-Q water, subsequently supplementing 0.074 g NaHCO_3,_ and adjusting the pH to 7.5 utilizing 0.1 M hydrochloric acid. The CA NPs were synthesized by addition of 4 mM exogenous CaCl_2_ (Ca^2+^) to freshly prepared 1 mL DMEM and incubation for 30 min at 37 °C.

To formulate and optimize the Fe/Mg-CA NPs, different concentrations of MgCl_2_ (Mg^2+^) (1, 2, 4, 6, 8, 10 and 12 mM) were added separately to two different mixtures, consisting of exogenous 4 mM Ca^2+^ and 1 µM Fe^3+^, and 4 mM Ca^2+^ and 5 µM Fe^3+^ in freshly prepared 1 mL DMEM, followed by 30 min of incubation at 37 °C. Afterward, 10% FBS (Fetal Bovine Serum) was added to each of the samples to hinder particle agglomeration.

### 2.3. Turbidity Measurement of Fe/Mg-CA NPs

The turbidity of CA formulated with 4 mM exogenous Ca^2+^ and Fe/Mg-CA NPs formulated with 4 mM Ca^2+^ and 1 µM Fe^3+^, and 4 mM Ca^2+^ and 5 µM Fe^3+^ with various concentrations of Mg^2+^ (1, 2, 4, 6, 8, 10 and 12 mM) was measured at 320 nm wavelength using 1800 MS UV-VIS spectrophotometer, Shimadzu, Japan. Each of the samples was prepared in triplicate, and the absorbance was calculated in terms of mean value and standard deviation.

### 2.4. Particle Size and Surface Charge Measurement of Fe/Mg-CA NPs

The average particle sizes of the samples were assessed using a Malvern nano zeta sizer (Malvern, Worcestershire, UK) and the zeta sizer software was employed for data analysis. Each of the samples was prepared in duplicate and the z-average diameter was estimated by dynamic light scattering (DLS), using a refractive index ratio of 1.325. The zeta potential was calculated in terms of mean value and standard deviation.

### 2.5. Optical Microscopic Imaging

An Olympus IX81 Fluorescence microscope (Shinjuku, Tokyo, Japan) was used to visualize the aggregation of the CA and the Fe/Mg-CA NPs formulated with 4 mM Ca^2+^, 1 µM Fe^3+^ and 4 mM Mg^2+^ (low Fe/Mg-CA), and 4 mM Ca^2+^, 5 µM Fe^3+^ and 12 mM Mg^2+^ (high Fe/Mg-CA). After incubation at 37 °C for 30 min, 10% FBS was added to each of the samples to hinder particle aggregation, and the respective samples of CA and Fe/Mg-CA NPs were placed into a 24-well plate and incubated at 37 °C for 60 min. Afterward, optical microscopic images were acquired immediately. The magnifications used were 4× (scale bar of 200 µm), 10× (scale bar of 50 µm), 20× and 40×.

### 2.6. Fourier Transform Infrared (FTIR) Spectroscopy

To determine the presence of carbonate (CO_3_^2−^) and phosphate (PO_4_^3−^) groups in the CA, low Fe/Mg-CA and high Fe/Mg-CA NPs, FTIR spectroscopy was performed utilizing a Varian 640-IR over the wavenumber range 4000–400 cm^−1^. Data were analyzed using the Varian Resolution Pro 640 software (Agilent, Santa Clara, CA, USA).

CA was formulated with 40 mM Ca^2+^, low Fe/Mg-CA NPs with 40 mM Ca^2+^, 10 µM Fe^3+^ and 40 mM Mg^2+^, and high Fe/Mg-CA with 40 mM Ca^2+^, 50 µM Fe^3+^ and 120 mM Mg^2+^ in bicarbonate buffered DMEM. The mixtures were then incubated at 37 °C for 30 min in order to induce the formation of the NPs. The samples were centrifuged twice at 3200 rpm for 30 min, and the supernatants were discarded. Small amounts of autoclaved Milli Q water were added to the pellets. The precipitated pellets were then kept at −20 °C for 30 min and lyophilized (Labconco, Kansas City, MO, USA) prior to the acquisition of IR spectra.

### 2.7. Stability Test of Fe/Mg-CA NPs

CA was synthesized by adding 4 mM exogenous Ca^2+^ to freshly prepared 1 mL DMEM at physiological pH 7.5, with subsequent incubation at 37 °C for 30 min. Low Fe/Mg-CA NPs were generated with 4 mM Ca^2+^, 1 µM Fe^3+^ and 4 mM Mg^2+^, and high Fe/Mg-CA with 4 mM Ca^2+^, 5 µM Fe^3+^ and 12 mM Mg^2+^ in freshly prepared 1 mL DMEM, followed by incubation at 37 °C for 30 min. Then 10% FBS was added to the prepared NPs. The turbidity was measured at 320 nm wavelength using an 1800 MS UV-VIS spectrophotometer, Shimadzu, Japan at 0, 4, 24, and 72 h to analyze the stability of the CA and Fe/Mg-CA NPs. Similarly, the z-average diameter of the samples was measured at 0, 4, 24, and 72 h using Malvern nano zeta sizer (Malvern, Worcestershire, UK) attached with a zeta sizer software for data analysis in order to evaluate the size as well as the level of aggregation and uniformity of the particles. Each of the samples was prepared in triplicate, and the turbidity was calculated in terms of mean value and standard deviation.

### 2.8. Field Emission Scanning Electron Microscopy (FESEM) and Energy Dispersive X-ray (EDX) Analysis

#### 2.8.1. FESEM Image Analysis for Fe/Mg- CA NPs in Absence of 10% FBS

FESEM was used to examine the morphological structure of CA, low Fe/Mg-CA and high Fe/Mg-CA NPs. Following incubation at 37 °C for 30 min, the respective mixtures of CA and Fe/Mg-CA NPs were centrifuged for 15 min at 13,000 rpm. The supernatant was removed from each sample, followed by the resuspension of the pellet in 1 mL of Milli Q water, and centrifugation at 13,000 rpm for another 15 min. The supernatants of the samples were decanted and the pellets were resuspended in 20 µL Milli Q water. Then 3 µL of the respective NP suspensions was dried at room temperature for 45 min on a glass slide and arranged on a specimen holder using carbon coating tape, followed by coating with Pt using a sputter coater with 30 mA sputter current for 40 s at 2.30 tooling factor. The samples were analyzed by scanning utilizing a focused fine beam of electrons. Finally, microscopic images of CA and both low and high Fe/Mg-CA nanocrystals were captured at 2 kV using FESEM (Hitachi/SU8010, Tokyo, Japan).

#### 2.8.2. FESEM Image Analysis for Fe/Mg-CA NPs in Presence of 10% FBS

CA, low Fe/Mg-CA and high Fe/Mg-CA NPs were prepared and incubated exactly in the same manner as indicated in Section 2.8.1, except for the addition of 10% FBS before centrifugation at 13,000 rpm for 15 min. The rest of the steps were the same as mentioned in Section 2.8.1 for FESEM analysis.

#### 2.8.3. FESEM Image Analysis for Fe-Carbonate Apatite (Fe-CA) and Mg-Carbonate Apatite (Mg-CA) NPs

To analyze the individual effects of Fe^3+^ and Mg^2+^ on CA, FESEM micrographs of Fe-CA and Mg-CA were captured. Fe-CA formulated with 4 mM exogenous Ca^2+^ and 5 μM exogenous Fe^3+^, and Mg-CA synthesized with 4 mM exogenous Ca^2+^ and 12 mM exogenous Mg^2+^ were prepared in bicarbonate buffered DMEM, followed by incubation at 37 °C for 30 min. Afterward, 10% FBS was added to each mixture, and they were centrifuged for 15 min at 13,000 rpm, followed by removal of the supernatant, resuspending of pellets in Milli Q water, and centrifuging again at 13,000 rpm for 15 min. 3 µL suspension of the respective NPs was dried on a glass slide for 40 min and arranged on a specimen holder using carbon-coated tape, followed by coating with Pt using a sputter coater with 30 mA sputter current for 40 s at 2.30 tooling factor.

#### 2.8.4. Energy Dispersive X-ray (EDX) Analysis

To identify the elements and their composition in selected sample regions, EDX (X-max, 50 mm, Horiba, Japan) analysis was carried out. The EDX analysis was performed for CA, low Fe/Mg-CA and high Fe/Mg-CA NPs. The analysis was carried out at a beam energy of 15 kV with an input count rate of 100,000. The data generated by the EDX represented the spectra concerning the peaks of the different elements present in the samples.

### 2.9. Flame Atomic Absorption Spectroscopy (FAAS)

CA NPs synthesized with 40 mM Ca^2+^, low Fe/Mg-CA NPs generated with 40 mM Ca^2+^, 10 µM Fe^3+^ and 40 mM Mg^2+^, and high Fe/Mg-CA formulated with 40 mM Ca^2+^, 50 µM Fe^3+^ and 120 mM Mg^2+^ were prepared in bicarbonate buffered DMEM through incubation at 37 °C for 30 min, followed by supplementation with 10% FBS. Afterward, the mixtures were centrifuged at 3200 rpm for 30 min. The supernatants were then discarded. A small volume of autoclaved Milli Q water was added to the pellets and then the samples were centrifuged again at 3200 rpm for 20 min. The pellets were then kept at −20 °C for 30 min and lyophilized (Labconco, Kansas City, MO, USA). Finally, the chemical composition (%) of Ca^2+^ in CA, and Ca^2+^, Fe^3+^, and Mg^2+^ in Fe/Mg-CA NPs were evaluated by FAAS.

The standard (gram/litre) for each element (Ca^2+^, Fe^3+^ and Mg^2+^) was prepared by dissolving the respective salts (CaCl_2_.2H_2_O, FeCl_3_.6H_2_O and MgCl_2_.6H_2_O) in 1N HCl acid. The lyophilized samples of CA, low Fe/Mg-CA and high Fe/Mg-CA were dissolved in 1N HCl mg/mL (g/L) concentration. The standard curves were used to detect the chemical composition (%) of Ca^2+^ in CA, and Ca^2+^, Fe^3+^, and Mg^2+^ in Fe/Mg-CA NPs. The amount of the sample intake for analysis was 4 to 5 mL/min. The measurement parameters used to plot the standard curve was demonstrated in Table 1.

The experiment was performed in duplicate, and the composition of Ca^2+^, Fe^3+^, and Mg^2+^ present in the samples was calculated in terms of mean value and standard deviation.

### 2.10. Binding Affinity of DOX with Fe/Mg-CA NPs

DOX-loaded CA (DOX-CA) was synthesized by adding different concentrations of DOX (1, 2.5, 5, 7.5 and 10 µM) and 20 mM Ca^2+^ into 1 mL of prepared DMEM at pH 7.5. Similarly, DOX-loaded low Fe/Mg-CA (DOX-low Fe/Mg-CA) and DOX-loaded high Fe/Mg-CA (DOX-high Fe/Mg-CA) were synthesized by adding different concentrations of DOX (1, 2.5, 5, 7.5 and 10 µM) to 20 mM Ca^2+^, 5 µM Fe^3+^ and 20 mM Mg^2+^, and 20 mM Ca^2+^, 25 µM Fe^3+^ and 60 mM Mg^2+^, respectively. After that, the mixtures were incubated at 37 °C for 30 min and centrifuged at 12,000 rpm for 20 min at 4 °C employing a benchtop micro-centrifuge. The supernatant was discarded and 10 mM EDTA in PBS was used to resuspend the pellets of the respective samples of DOX-CA and DOX-Fe/Mg-CA NPs. The samples were then placed in a 96 well-plate and subjected to 2030 multilabel reader victor TMX5 containing the PerkinElmer 2030 manager software (PerkinElmer, Waltham, MA, USA).

The fluorescence intensity of the bound drugs was measured at an excitation and emission spectra of 485 nm and 535 nm, respectively. The standard curve was obtained by plotting the fluorescence intensity against the concentration of the free DOX. The amounts of DOX in DOX-CA, DOX-low Fe/Mg-CA and DOX-high Fe/Mg-CA NPs were estimated utilizing the standard curve. The binding percentage of DOX to CA and Fe/Mg-CA NPs were calculated using the following formula—
$$DOX\;Binding\;\left(\%\right)=\frac{B}{A}\times 100\%$$
where,A = total concentrations of DOX used in the experiment (i.e., 1, 2.5, 5, 7.5 and 10 µM)B = different concentrations of DOX (i.e., 1, 2.5, 5, 7.5 and 10 µM) bound NPs in the pellet resuspended with 10 mM EDTA in PBS.

The experiment was performed in triplicate, and the results were analyzed in terms of mean value and standard deviation.

### 2.11. pH Dissolution Study of Fe/Mg-CA NPs

CA NPs generated with 20 mM Ca^2+^, low Fe/Mg-CA NPs fabricated with 20 mM Ca^2+^, 5 µM Fe^3+^ and 20 mM Mg^2+^, and high Fe/Mg-CA synthesized with 20 mM Ca^2+^, 25 µM Fe^3+^ and 60 mM Mg^2+^ were added to 200 µL freshly prepared DMEM media, prior to incubation at 37 °C for 30 min. After that, 800 µL DMEM media of different pHs (7.5, 7, 6.5, 6 and 5.5) was supplemented to each sample. The turbidity of individual samples was measured at 320 nm wavelength using 1800 MS UV-VIS spectrophotometer (Shimadzu, Japan). Each of the samples was prepared in triplicate, and the turbidity was calculated in terms of mean value and standard deviation.

### 2.12. pH Dependent Release of DOX from Fe/Mg-CA NPs

DOX-CA was synthesized by adding 5 µM DOX to 20 mM Ca^2+^ in 200 µL freshly prepared DMEM media. Similarly, DOX-low Fe/Mg-CA was fabricated by adding 5 µM DOX to 20 mM Ca^2+^, 5 µM Fe^3+^ and 20 mM Mg^2+^ and DOX-high Fe/Mg-CA was generated with 20 mM Ca^2+^, 25 µM Fe^3+^ and 60 mM Mg^2+^ in 200 µL DMEM media. The mixtures were then incubated at 37 °C for 30 min. Followed by incubation, the samples were suspended in 800 µL DMEM media with different pHs (7.5, 7, 6.5, 6 and 5.5), and centrifuged at 12,000 rpm for 20 min at 4 °C by using a benchtop micro-centrifuge. The supernatant was then discarded, and 10 mM EDTA in PBS was used to dissolve the pellets of DOX-CA, DOX-low Fe/Mg-CA and DOX-high Fe/Mg-CA NPs for active release of DOX from the samples. The fluorescence intensity of the bound drugs was measured at an excitation wavelength of 485 nm and an emission wavelength of 535 nm using the 2030 multilabel reader victor X5 attached with PerkinElmer 2030 manager software (PerkinElmer, Waltham, Ma, USA). The standard curve was used to calculate the binding of DOX with CA, low Fe/Mg-CA and high Fe/Mg-CA at different pH. Data were represented as the percentage of DOX released from the NPs, calculated by utilizing the following expression—
$$DOX\;Released\;from\;NPs\;\left(\%\right)=\frac{A-B}{A}\times 100\%$$
where,A = NP bound drug at pH 7.5B = NP bound drug at different pHs (7.5, 7, 6.5, 6 and 5.5)

The experiment was performed in triplicate, and the findings were analyzed in terms of mean value and standard deviation.

### 2.13. Cell Culture and Seeding

MCF-7 (Michigan Cancer Foundation-7) is the human mammary tumor cell line used for cell culture and seeding. MCF-7 cells were cultured in 25 cm^2^ flasks in complete DMEM (CDMEM) media containing 10% FBS, 1% penicillin and streptomycin antibiotic having pH 7.4. The flasks were then incubated at 37 °C in a humidified environment containing 5% CO_2_. Rapidly growing MCF-7 cells were trypsinized, followed by washing through centrifugation, and seeding in 24-well plates (Greiner, Frickenhause, Germany) where each well contained 50,000 cells. The 24-well plates were then incubated overnight.

### 2.14. Cell Treatment and Preparation of Free DOX, DOX-CA and DOX-Fe/Mg-CA NPs

DOX-loaded low Fe/Mg-CA NPs were formulated by adding 4 mM Ca^2+^, 1 µM Fe^3+^ and 4 mM Mg^2+^, and DOX-loaded high Fe/Mg-CA NPs were synthesized by adding 4 mM Ca^2+^, 5 µM Fe^3+^ and 12 mM Mg^2+^,with different concentrations of DOX (1 nM, 10 nM, 100 nM, and 1 μM) in 1 mL freshly prepared filtered bicarbonate-buffered DMEM (pH 7.4). Similarly, DOX-loaded CA NPs were prepared by adding 4 mM Ca^2+^ to different concentrations of DOX (1 nM, 10 nM, 100 nM, and 1 μM) in 1 mL bicarbonate-buffered DMEM. The mixtures were then incubated at 37 °C for 30 min, followed by the addition of 10% FBS. Free DOX of desired concentrations (1 nM, 10 nM, 100 nM, and 1 μM) was added to DMEM prior to the incubation and supplementation with 10% FBS. Low Fe/Mg-CA, high Fe/Mg-CA, and CA NPs were fabricated in bicarbonate buffered DMEM, incubated at 37 °C for 30 min and supplemented with 10% FBS. After that, the old DMEM from each well was swapped with the medium containing the prepared mixtures. The mixtures were then incubated for two consecutive days at 37 °C maintaining a humidified environment comprising 5% CO_2_.

### 2.15. Cell Viability by MTT (3-4,5-Dimethylthiazol-2-yl-2,5-Diphenyltetrazolium Bromide) Assay

To assess the cell viability by MTT assay, 50 μL of MTT (5 mg/mL in PBS) was added to each well in the 24-well plate to facilitate the development of formazan crystals. After that, the 24-well plate was incubated at 37 °C for 4 h. The media having MTT solution was then removed from each well, and 300 μL of DMSO solution was subsequently added and mixed by pipetting to dissolve the purple formazan crystals. A spectrophotometric microplate reader (BIO-RAD-Microplate Reader, Hercules, CA, USA) was used to measure the absorbance of the samples at an optical/reference wavelength of 595/630 nm. Each of the samples was prepared in triplicate, and the absorbance was calculated in terms of mean value and standard deviation. The cell viability (%) was calculated from the following formula—
$$Cell\;Viability\;\left(\%\right)=\frac{Absorbance_{NP\;bound\;drug}-Absorbance_{free\;drug}}{Absorbance_{NP}-Absorbance_{free\;drug}}\times 100\%$$

The cytotoxicity enhancement (%) was calculated by the following formula—
Cytotoxicity Enhancement (%) = (100 − CV_NP bound drug_) − (Cytotoxicity_NP_ + Cytotoxicity_DOX_)
where,CV_NP bound drug_ = Cell Viability of NP Bound DrugCytotoxicity_NP_ = 100 − Cell Viability of NPCytotoxicity_DOX_ = 100 − Cell Viability of DOX

### 2.16. Cellular Uptake of DOX-Loaded Fe/Mg-CA NPs in MCF-7 Cells using Fluorescence Microscopy

MCF-7 cells were seeded in 24-well plates containing 50,000 cells per well and incubated for 24 h at 37 °C and 5% CO_2_. The wells were then treated with free DOX, DOX-CA formulated with 4 mM Ca^2+^ and 5 µM DOX, DOX-low Fe/Mg-CA generated with 4 mM Ca^2+^,1 µM Fe^3+^, 4 mM Mg^2+^ and 5 µM DOX, and DOX-high Fe/Mg-CA synthesized with 4 mM Ca^2+^, 5 µM Fe^3+^, 12 mM Mg^2+^ and 5 µM DOX, followed by incubation and the addition of 10% FBS. At 4 h and 24 h, the supernatant of the culture media was discarded. The cells were then rinsed with 100 µL of 10 mM EDTA in PBS (phosphate-buffered saline) to eliminate the extracellular particles, prior to washing twice with 100 µL PBS. After that, 100 µL PBS was added to each well and the images of the cells were acquired using the Olympus Fluorescence microscope IX81 (Shinjuku, Tokyo, Japan) attached with a CellSens Dimension software.

### 2.17. Quantitative Analysis of Cellular Uptake for DOX-Loaded Fe/Mg-CA NPs

After acquiring fluorescence micrographs, the PBS from each well was discarded, and 100 µL Lysis buffer was supplemented to each well to facilitate cell lysis. The fluorescence intensity was then evaluated at an excitation wavelength of 485 nm and an emission wavelength of 535 nm by means of a 2030 multilabel reader victor X5 (PerkinElmer, Waltham, MA, USA) attached with the PerkinElmer 2030 manager software. The DOX concentration was evaluated utilizing the standard curve. The cellular uptake (%) was measured utilizing the following formula—
$$Cellular\;Uptake\;\left(\%\right)=\frac{F_{Internalized\;Drug}}{F_{Free\;Drug}}\times 100\%$$
where,F_Free Drug_ = Initial fluorescence intensity of free DOXF_Internalized drug_ = Fluorescence intensity of the amount of DOX internalized from the respective NPs into the cells

Each of the samples was prepared in triplicate, and the fluorescence intensity was calculated in terms of mean value and standard deviation.

### 2.18. Protein Corona Analysis

#### 2.18.1. In-Solution Digestion of CA and Fe/Mg-CA Protein Corona for Mass Spectrometric Analysis

Particles of CA were formulated with 4 mM exogenous Ca^2+^, whereas low Fe/Mg-CA and high Fe/Mg-CA were generated, respectively, with 4 mM Ca^2+^, 1 µM Fe^3+^ and 4 mM Mg^2+^, and with 4 mM Ca^2+^, 5 µM Fe^3+^ and 12 mM Mg^2+^. After adding 10% FBS, the particles were incubated for 20 min at 37 °C. The particle suspensions were then centrifuged at 13,000 rpm for 15 min, followed by removal of the supernatants and washing of the pellets in Milli Q water. The process of centrifuging and discarding the supernatant was repeated. The pellets were dissolved in 100 µL of 50 mM EDTA in H_2_O. After that, 25 µL of 100 mM (NH_4_)_2_CO_3_ solution, 25 µL tetrafluoroethylene (TFE) denaturing agent and 1 µL of 200 mM dithiothreitol (DTT) solution were mixed to the protein mixture. Following this, the samples were vortexed and heated using a heating block at 60 °C for 1 h. 4 µL of 200 mM iodoacetamide (IAM) was added to the protein mixture and vortexed. Subsequently, the mixture was incubated for 1 h in the dark at room temperature in the presence of 100 µL (NH_4_)_2_CO_3_ solution (100 mM). The mixture was then digested overnight with MS Grade 25 µL of Trypsin (1 µg/mL) at 37 °C. Lastly, the reaction was halted with 1 µL of formic acid, and the samples were speed vacuumed overnight prior to analysis by Q-TOF LC-MS/MS.

#### 2.18.2. Sample Preparation for Mass Spectrometry-Based Proteomics

10 µL of 0.1% formic acid solution was applied to the dry peptide digest and sonicated for 10 min, using ice to maintain an ambient temperature. Samples were then centrifuged (14,000× *g*, 5 min). Subsequently, 5 µL of supernatant was kept in MS tube and directly moved to LC-QTOF auto-sampler before analysis.

#### 2.18.3. High Efficiency Nanoflow Liquid Chromatography Electrospray-Ionization Coupled with Mass Spectrometry

The columns were equilibrated by loading the peptide digests into an Agilent Poroshell 300 Å pore C18 columns (Agilent, Santa Clara, USA) using a 0.1% formic acid. Elution of peptides was carried out using 90% acetonitrile in 0.1% formic acid (solution B), gradients of 5% solution B over 0–30 min and 75% solution B over 30–39 min. The positive polarity of quadrupole-time of flight (Q-TOF) and 5 L/min of gas flow at 325 °C were fixed to operate the mass spectra, maintaining a fragmented voltage of 1750 V and 360 V, respectively. Auto MS mode was used to assess the peptide spectrums using the range of 110–3000 m/z for MS scan and 50–3000 m/z for MS/MS scan. Acquisition rates of 2 (spectra/s) and 4 (spectra/s) were employed for MS and MS/MS, respectively. The spectrums were then studied with Agilent MassHunter (Agilent Technologies, Santa Clara, CA, USA) data acquisition software and data were analyzed with PEAKS 8.0 software (Bioinformatics Solutions Inc., Waterloo, ON, Canada).

#### 2.18.4. Protein Quantification and Identification through Automated De Novo Sequencing (PEAKS Studio 8.0)

For protein identification, database search (SwissProt.Mus_musculus) with de novo sequencing was conducted. A homology search was performed using PEAKS Studio 8.0 (Bioinformatics Solution Inc., Waterloo, ON, Canada). Carbamidomethylation modification was done with maximum mixed cleavages at 3. Fragment and parent mass error tolerance were both set to 0.1 Da with monoisotopic mass as the precursor mass search type. Digestion was carried out with Trypsin. Incorrect proteins were filtered by using a false discovery rate (FDR) of 1% and unique peptides ≥ 1. A -10lgP score of greater than 20 indicated that detected proteins were reliable.

With the aid of PEAKS Q protein quantification software, relative differential changes of proteins present in CA, low Fe/Mg-CA and high Fe/Mg-CA were quantified with a label-free quantification method based on the relative intensities of peptide ion peaks. Feature detection of the respective samples was executed using EM (expectation-maximization) algorithm. A high-performance retention time alignment algorithm was used to align the features of the same peptide from separate samples. To differentiate the groups among NPs and the intensity of a quantifiable peptide, the groups were color-coded in the heat map summary. The importance of the peptide was indicated by its -10LogP score. The cut off value was set at 20 which was significant to the *p*-value of 0.01. The value of 1 represented a 2-fold change by threshold ratio. Filtered proteins were displayed on the Heat Map for quantitative analysis. The heat map of the representative proteins depicts the relative protein abundance of each protein group. If the representative proteins have similar expressions, an aggregated trend was observed across the samples. The hierarchical clustering was produced by using the neighbor-joining algorithm with a Euclidean distance similar to the measurement of the log 2 ratios of abundance in relation to the average abundance.

### 2.19. Animal Biodistribution Study

The biodistribution study was carried out according to the procedure permitted by the Monash Animal Ethics Committee (MARP/2016/126). Female Balb/c mice (6 to 8 weeks old with a bodyweight of around 20–25 g, maintained in 12:12 light:dark condition, and provided with ad libitum and water), obtained from School of Medicine and Health Science Animal Facility, Monash University, were subcutaneously injected on the mammary pad with 4T1 cells (1 × 10^5^/mouse) in 180 µL PBS. The tumor outgrowth was monitored regularly by touching the injected section through the index finger. When the tumor volume reached around 75 mm^3^, mice were injected with DOX solution, DOX-loaded CA NPs, DOX-loaded low Fe/Mg-CA NPs and DOX-loaded high Fe/Mg-CA NPs (DOX dose: 5 mg/kg) through tail vein injection. After 24 h of treatment, mice were euthanized through cervical dislocation to collect tumor and other major organs such as the brain, liver, spleen, kidney, heart, muscle and lungs. The tumor and the organs were washed in PBS, followed by the addition of 500 µL Lysis buffer containing sodium fluoride and stored at −150 °C until further analysis.

Later, the organs were lysed using a mechanical homogenizer, centrifuged at 8000 rpm for 20 min at 4 °C. Then 200 µL of the supernatant was placed in a 96-well opti-plate (Nunc) to measure the fluorescence intensity at an excitation/emission wavelength of 485/535 nm using a 2030 multilabel reader victor X5 (PerkinElmer, Waltham, MA, USA) attached with the PerkinElmer 2030 manager software. The values were blank corrected with respect to the untreated group of mice for each organ and expressed as mean ± SEM of fluorescence intensity/500mg of tissue mass.

### 2.20. Blood Serum Analysis

The amount of DOX present in blood serum was studied by collecting blood from all mice (n = 3 mice per group). After 24 h of treatment, blood was collected through cardiac puncture utilizing 1 mL syringe with a 25 g (gauge) needle, followed by an auxiliary approach of euthanasia to decease the animal. The serum from the blood was collected by allowing the blood to clot at ambient temperature, prior to centrifugation at 10,000 rpm at 4 °C for 20 min. Then 200 µL of the supernatant (serum) from each sample was collected in a 96-well opti-plate (Nunc) and fluorescence intensity was measured at an excitation/emission wavelength of 485/535 nm using a 2030 multilabel reader victor X5 (PerkinElmer, Waltham, MA, USA) attached with the PerkinElmer 2030 manager software. The data were presented as mean ± standard deviation.

## 3. Results and Discussion

### 3.1. Measurement of Turbidity as a Reflection of Particle Formation

CA, low Fe/Mg-CA and high Fe/Mg-CA NPs were fabricated following the procedure referred to in the Materials and Method section. The particles of CA were generated through the chemical reaction between the positively charged Ca^2+^ and the negatively charged PO_4_^3−^ and CO_3_^2−^ under slightly alkaline pH (7.5) and physiological temperature (37 °C). Similarly, low Fe/Mg-CA and high Fe/Mg-CA particles were synthesized by the inclusion of two additional cations, Mg^2+^ and Fe^3+^, which could partially substitute Ca^2+^ in the apatite.

Fabrication of Fe/Mg-CA NPs was verified through turbidity analysis at 320 nm. It was observed that with increasing concentrations of Mg^2+^ (1, 2, 4, 6, 8, 10 and 12 mM), the turbidity of the respective samples of low Fe/Mg-CA and high Fe/Mg-CA NPs decreased almost proportionately (Figure 1). This might be due to the notion that Mg^2+^ inhibited the growth of particles as one of the reactants by partially replacing the Ca^2+^ ions and distorting the lattice structure of the apatite NPs, and, thus, generating small crystals in a dose-dependent manner [39,44]. However, with Fe-CA NPs prepared with low and high Fe amounts, but without Mg, high growth of particles was observed with turbidity values of nearly 0.35 and 0.48, respectively (Figure 1). This might be due to the presence of Fe^3+^ as one of the reactants which induced the growth of the particles, resulting in the formation of aggregates.

### 3.2. Particle Size and Surface Charge Measurement

For effective drug delivery, NPs with the range of 100–1000 nm were often used [8]. This is because blood vessels are frequently inflated and fenestrated having an average pore diameter of 380–400 nm [46] in a fast-growing tumor. Therefore, particles within the nano-size range could accumulate in solid tumors by means of the enhanced permeability and retention (EPR) effect after passing through the leaky endothelial tissue of the tumor vasculature. The capillaries of normal tissues contain tight junctions (~10 nm) that restrict diffusion of nano-sized particles [47,48]. In addition, if carriers are too big in size, they can clog the blood vessels. Moreover, excessively small (˂8 nm) or large (200–1000 nm) NPs are rapidly cleared by the kidney and reticuloendothelial systems, respectively.

The z-average diameters of Fe/Mg-CA NPs formed with increasing concentrations of Mg^2+^ were measured using Malvern Nano Zetasizer. Both low and high Fe/Mg-CA particles revealed a decreasing pattern in the z-average diameter with increasing concentrations of Mg^2+^. The Fe-CA particles formed without any exogenous Mg^2+^ (0 mM) were bigger than CA particles, with higher Fe^3+^ (5 µM) concentration inducing the formation of larger particles than the lower Fe^3+^ (1 µM) one (Figure 2), which was in-line with the significant difference in turbidity observed for the two formulations (Figure 1). In both cases (low Fe-CA and high Fe-CA), the inclusion of 2 mM of Mg^2+^ caused a slight decrease in particle diameter, while 4 mM of Mg^2+^ led to a sharp decrease particularly in the size of low Fe/Mg-CA particles, generating particles of 389 ± 19 nm. On the other hand, high Fe/Mg-CA particles revealed a z-average diameter of 654 ± 11 nm with 4 mM Mg^2+^ concentration. However, high Fe/Mg-CA showed markedly small sized particles at 12 mM (189 ± 9 nm) Mg^2+^ concentration (Figure 2).

In general, the z-average diameters of Fe-Mg-CA NPs with increasing concentrations of Mg^2+^ was dose-dependent. These findings imply that Mg^2+^ along with FBS inhibited the growth of the particles and prevented self-aggregation between them.

Depending on the results of turbidity and z-average diameter, two formulations of Fe/Mg-CA NPs formed by exogenously adding 4 mM Ca^2+^, 1 µM Fe^3+^ and 4 mM Mg^2+^ (low Fe/Mg-CA), and 4 mM Ca^2+^, 5 µM Fe^3+^ and 12 mM Mg^2+^ (high Fe/Mg-CA) were selected for further analysis.

The size distribution of CA, low Fe/Mg-CA and high Fe/Mg-CA NPs was analyzed based on intensity and volume (Appendix A), indicating that Fe/Mg-CA particles were apparently smaller and more homogeneous than CA ones. It should be noted that Fe/Mg-CA NPs (high Fe/Mg-CA and low Fe/Mg-CA) following 5-fold dilution of the original suspension resulted in much smaller particles with lower PDI (polydispersity index) values than CA NPs (Appendix A), suggesting that particles in high concentration probably formed transient aggregates which could be separated by dispersion into a bigger space.

As the electric charge on the surface of NPs would determine physical stability and biocompatibility, zeta potential (ζ) of CA and Fe/Mg-CA suspensions were examined. The surface charge of CA was less electronegative than that of low Fe/Mg-CA and high Fe/Mg-CA NPs. CA had a zeta potential of nearly −11 mV [44], whereas low Fe/Mg-CA and high Fe/Mg-CA showed a zeta potential of around −12 mV and −15 mV, respectively (Figure 3). The more electronegativity of Fe/Mg-CA NPs suggests that anions from the solution were probably accumulated on cation-rich (Ca/Fe/Mg) domains through electrostatic interactions. When Fe^3+^ and Mg^2+^ ions were added, some of the Ca^2+^ ions in the NPs could be substituted by the Mg^2+^ and Fe^3+^ ions (Figure 3).

### 3.3. Observation of Particle Aggregates under Optical Microscope

Optical micrographs of CA, low Fe/Mg-CA and high Fe/Mg-CA NPs were captured to verify the aggregation of the particles (Appendix A). Compared to CA NPs, Fe/Mg-CA NPs exhibited fewer aggregates (Appendix A). High Fe/Mg-CA NPs showed minimal aggregation (Appendix A) due to the incorporation of a high amount of Mg^2+^, inhibiting the growth and distorting the lattice structure of the apatite NPs.

### 3.4. Characterization of Fe/Mg-CA NPs by Fourier Transform Infrared (FTIR) Spectroscopy

The formation of CA, low Fe/Mg-CA, and high Fe/Mg-CA was confirmed through the FTIR spectra (Appendix A). The FTIR spectra showed the formation of CA particles as reflected by the peaks for carbonate (CO_3_^2−^) and phosphate (PO_4_^3−^) in the respective samples (Table 2). The absorption bands at 1410–1670 cm^−1^ and 880 cm^−1^ were attributed to CO_3_^2−^. The bands for PO_4_^3−^ were assigned at 1000–1100 cm^−1^ and 550–650 cm^−1^ [49]. The slight shifting of the peaks for CO_3_^2^^−^ and PO_4_^3^^−^ for Fe/Mg-CA NPs compared to CA NPs might be due to the interactions of the Fe^3+^ and Mg^2+^ with the CO_3_^2^^−^ and PO_4_^3^^−^ in the crystal lattice. The FTIR spectra for CA and Fe/Mg-CA NPs were presented in Table 2.

### 3.5. Stability Analysis of Fe/Mg-CA NPs through Turbidity Measurement and Z-Average Diameter

Stability analyses based on turbidity measurement at 320 nm and z-average diameter were carried out to have an idea on the extent of particle aggregation that might affect the half-life of the particles and their biocompatibility following intravenous administration. An increasing trend in turbidity was observed for CA, low Fe/Mg-CA, and high Fe/Mg-CA NPs with increasing incubation time up to 72 h (Figure 4). However, the overall turbidity of Fe/Mg-CA NPs was less compared to that of CA NPs, which might be due to less aggregation of Fe/Mg-CA NPs compared to CA NPs, thus resulting in small size of the particles (Figure 5) over the period of 72 h. No significant change was observed in the z-average diameter for CA over a period of 72 h (Figure 5). However, a small reduction in PDI value (Appendix A) was noticed. A slight increase in the z-average diameter with a reduction in the PDI value was observed for high Fe/Mg-CA NPs over a period of 72 h. Low PDI value is regarded as an indicator of less aggregated and monodispersed particles. The synergistic effect of Mg^2+^ and FBS which inhibited the growth of the particles and reduced their self-aggregation might help lower the PDI values for Fe/Mg-CA NPs. The size distribution of CA and Fe/Mg-CA NPs over the same time period based on volume was represented in Appendix A.

### 3.6. Characterization through FESEM and EDX

#### 3.6.1. Morphological Analysis of Fe/Mg-CA NPs through FESEM in Absence of 10% FBS

FESEM images were captured in order to understand the morphological structure of the NPs, revealing that CA, low Fe/Mg-CA and high Fe/Mg-CA nano-crystals in absence of 10% FBS were nearly spherical with rough surface morphology (Figure 6). On close observation (500 nm and 100 nm scale range), the surface of Fe/Mg-CA NPs was found slightly rougher than that of CA (Figure 6), which might be due to the effect of Fe^3+^ on the crystal structure. The average particle size for low Fe/Mg-CA was between 100–200 nm, and for high Fe/Mg-CA was between 80–150 nm, which was smaller compared to the size of CA (150–250 nm). The smaller size of the Fe/Mg-CA NPs was due to the addition of exogenous Mg^2+^ which inhibited the growth of the particles by distorting the lattice structure through partial replacement of the Ca^2+^ ions present in the apatite. However, the size of the particles obtained from FESEM analysis was a bit smaller (±100 nm) in relation to the z-average diameter acquired from the Zetasizer.

#### 3.6.2. Morphological Analysis of Fe/Mg-CA NPs through FESEM in Presence of 10% FBS

The FESEM micrographs showed that all three different particles have a spherical shape in the presence of 10% FBS. However, with the addition of FBS, the z-average diameter of CA decreased from almost 150–250 nm to 170–130 nm (Figure 7A). In the same way, low Fe/Mg-CA and high Fe/Mg-CA NPs displayed a reduction in z-average diameter from approximately 100–200 nm to 40–140 nm (Figure 7B), and 80–150 nm to 20–50 nm, respectively, with exception of few particles having a z-average diameter within the range of 100–200 nm (Figure 7C).

#### 3.6.3. Individual effects of Fe^3+^ and Mg^2+^ on CA NPs by FESEM Image Analysis

To determine the individual effect of Fe^3+^ and Mg^2+^ on CA NPs, FESEM images of Fe-CA and Mg-CA were captured. It was apparent that both Fe-CA and Mg-CA showed spherical geometry. The presence of Fe^3+^ as one of the reactants induced growth and aggregation of the Fe-CA particles with rough surface morphology (Figure 8A). On the contrary, Mg^2+^ inhibited the growth of the Mg-CA particles and drastically reduced the size to approximately 22 nm (Figure 8B) compared to Fe-CA NPs (250 nm) (Figure 8A). Hence, the addition of Mg^2+^ resulted in small, uniform particles (Figure 8B).

#### 3.6.4. Elemental Analysis by Energy Dispersive X-ray (EDX)

EDX analysis of both low and high Fe/Mg-CA NPs confirmed the presence of Ca, Mg, P, C, and O, although the presence of Fe in Fe/Mg-CA NPs was not detected (Appendix A). This could be due to the presence of trace amount (µM concentration) of Fe, unlike Ca and Mg which were present in mM concentration. EDX has an overall sensitivity and minimal detection limit of approximately 0.1 wt. (%) for all elements. Hence, it failed to detect Fe. The presence of a large Si peak in the spectrum was due to the glass (SiO_2_) plate used for the analysis. Pt was detected, as the samples were coated with Pt for the analysis. Na was identified as DMEM used for sample preparation contained both endogenous and exogenous Na salts. The table below shows the Ca/P and Ca+Mg/P ratio wt. (%) of CA and Fe/Mg-CA NPs.

The Ca/P molar ratio for CA was 2.00 ± 0.13 which was higher (Table 3) than the established stoichiometric ratio of HA (1.67) [50]. This could be explained with the notion that CA might have entered the phase of tetracalcium phosphate (TTCP), Ca_4_O(PO_4_)_2_ which contained a Ca/P molar ratio of 2.00 [43]. However, it was reported that the Ca/P molar ratio for CA could be up to 3.33 [51]. The Ca/P molar ratio for low Fe/Mg-CA (3.65 ± 0.11) and high Fe/Mg-CA (4.27 ± 0.22) NPs were greater than the stoichiometric ratio of HA (1.67), suggesting that interactions of Fe and Mg within the apatite structure drastically influence the elemental composition of the resultant particles.

The amount of Mg^2+^ substituted for Ca^2+^ in low Fe/Mg-CA and high Fe/Mg-CA was higher compared to CA because CA only contained 0.81 mM of endogenous Mg^2+^ whereas Fe/Mg-CA NPs contained both endogenous and exogenous Mg^2+^, indicating that more Mg^2+^ was substituted for Ca^2+^ in Fe/Mg-CA relative to CA NPs. Table 4 shows the amounts of Mg^2+^ substituted in Fe/Mg-CA and CA NPs.

### 3.7. Elemental Analysis by Flame Atomic Absorption Spectroscopy (FAAS)

The standard curves for Ca, Fe and Mg were shown in Appendix A. It was observed that the weight (%) of Ca^2+^ in low Fe/Mg-CA was slightly lower compared to high Fe/Mg-CA and CA. However, the weight (%) of Mg^2+^, as well as Fe^3+^, was greater in high Fe/Mg-CA than in low Fe/Mg-CA. This might be due to the incorporation of the increased amounts of Mg^2+^ and Fe^3+^ in high Fe/Mg-CA. Table 5 shows the concentrations of Ca^2+^, Fe^3+^, and Mg^2+^ in weight (%).

### 3.8. Binding Affinity of DOX Towards Fe/Mg-CA NPs

Encapsulation of drugs with NPs could enhance therapeutic effectiveness by improving the stability, solubility and targeted delivery of the drugs to tumor cells [52]. Additionally, drug-loaded NPs could play a vital role in enhancing cellular internalization via endocytosis [52].

In order to determine the binding affinity of DOX with low and high Fe/Mg-CA NPs, the binding efficiency of Fe/Mg-CA NPs with increasing concentrations of DOX (1, 2.5, 5, 7.5 and 10 μM) was evaluated (Figure 9). The standard curve of DOX was presented in Appendix A.

The loading capacity of CA for DOX showed an increasing trend with increasing concentrations of the drug (Figure 9). The results revealed that at 1, 2.5, 5, 7.5 and 10 µM concentrations of DOX, the binding affinity of DOX for CA was 7% ± 2, 12% ± 2, 20% ± 1, 25% ± 1 and 30% ± 3, respectively, which was in-line with our previous studies which displayed 19% binding affinity of DOX for CA at 5 µM concentration [53], and 26.25% at 10µM concentration [44].

The amount of DOX bound to both low and high Fe/Mg-CA NPs was increased up to 5 µM DOX concentration. Above 5 µM DOX, no significant change in the binding affinity of DOX to Fe/Mg-CA NPs (around 30%) was observed (Figure 9). This might be due to the saturation of DOX binding sites for Fe/Mg-CA NPs. However, the overall binding affinity of DOX with both low and high Fe/Mg-CA NPs was higher compared to CA NPs. The complexation of DOX with CA and Fe/Mg-CA NPs was due to the electrostatic interactions between the positively charged DOX and the negatively charged CO_3_^2−^ and PO_4_^3−^ of CA and Fe/Mg-CA NPs.

### 3.9. In Vitro pH-Dependent Dissolution Study of Fe/Mg-CA NPs

Tumor cells exhibited acidic pH due to an increased level of glycolysis [54]. To predict drug release from both low and high Fe/Mg-CA NPs in the tumor tissue environment, the degradability of Fe/Mg-CA NPs in acidic pH was investigated through turbidity measurement at 320 nm. With decreasing pH, a gradual reduction in turbidity was observed for CA, low Fe/Mg-CA and high Fe/Mg-CA NPs (Figure 10), indicating that CA and Fe/Mg-CA NPs could disintegrate in endosomal acidic pH, but remain stable in the physiological pH (pH 7.4).

### 3.10. pH-Dependent DOX Release Kinetics from DOX-Loaded Fe/Mg-CA NPs

The pH-dependent release of drugs from NPs based on the acidic pH of the tumor microenvironment and intracellular compartments (endosomes or lysosomes) of individual cancer cells could promote the build-up of the free drugs inside the cancer cells. Moreover, the level of toxicity towards healthy cells could be mitigated and multidrug resistance could be hindered [32,44,55,56,57]. To quantitate pH-dependent drug release, the release profiles of DOX from DOX-loaded CA, and DOX-loaded low and high Fe/Mg-CA NPs were monitored at both physiological pH (7.5), and acidic pH (5.5–6.5). The amounts of DOX released from both low Fe/Mg-CA and high Fe/Mg-CA NPs were noticeably lower at the blood pH (Figure 11). However, with decreasing pH, the release of DOX from CA, low Fe/Mg-CA and high Fe/Mg-CA NPs was significantly increased. At pH 7.5, the amount of DOX released from CA, low Fe/Mg-CA and high Fe/Mg-CA were 3% ± 1, 4% ± 3 and 3% ± 2, respectively, whereas nearly 80% of DOX was released from CA, low and high Fe/Mg-CA NPs at pH 5.5 (Figure 11). The rapid drug release at acidic pH might be attributed to the fast dissolution rates of CA and Fe/Mg-CA NPs which could consume H^+^ ions through their PO_4_^3−^ and CO_3_^2−^ groups.

### 3.11. Cell Viability Assessment by MTT Assay

MTT colorimetric assay was used to determine the effects of NPs and drug-loaded NPs on cell proliferation and metabolic activities of cells [58]. The cytotoxic effects of CA, low and high Fe/Mg-CA, DOX-loaded CA, DOX-loaded low and high Fe/Mg-CA NPs in relation to free DOX were determined against MCF-7 cells. The cell viability of CA and both low and high Fe/Mg-CA NPs was around 89%, suggesting negligible inherent toxicity of the NPs. The slight toxicity of the particles was apparently due to the excessive deposition of the respective NPs on the surface of the MCF-7 cells. As shown in Figure 12, DOX-loaded CA, DOX-loaded low Fe/Mg-CA and DOX-loaded high Fe/Mg-CA exhibited more potent cytotoxic effects compared to free DOX. Furthermore, DOX-loaded NPs formed with increased concentrations of DOX (1 nM, 10 nM, 100 nM, and 1 µM) showed enhanced cell growth inhibition compared to free DOX. However, DOX-loaded CA demonstrated a slight rise in cell viability percentage compared to DOX-loaded low Fe/Mg-CA and high Fe/Mg-CA NPs (Figure 12).

The cytotoxicity enhancement was further evaluated for DOX-loaded CA, DOX-loaded low Fe/Mg-CA and DOX-loaded high Fe/Mg-CA NPs Appendix A). It was evident that the cytotoxicity enhancement in DOX-loaded low and high Fe/Mg-CA NPs was slightly greater than DOX-loaded CA NPs. The most prominent effect was observed at 1 μM DOX concentration for DOX-loaded CA, and at 100 nM and 1 μM DOX concentrations for DOX-loaded low and high Fe/Mg-CA NPs (Appendix A). This enhancement in cytotoxicity could be described with the notion that the small size of Fe/Mg-CA NPs could lead to higher cellular internalization of the loaded drugs compared to CA NPs, resulting in the accumulation of a sufficient amount of free drugs in the cytosol for effective cell killing.

### 3.12. In Vitro Cellular Uptake of DOX-Loaded Fe/Mg-CA NPs in MCF-7 Cells

In order to examine the ability of low Fe/Mg-CA and high Fe/Mg-CA NPs in promoting the accumulation of DOX in MCF-7 cells after 4 h and 24 h of treatment, the cellular uptake of DOX-loaded low and high Fe/Mg-CA NPs was evaluated by fluorescence microscopy. It was observed that after 4 h of treatment, the fluorescence intensity of DOX-loaded CA, low and high Fe/Mg-CA NPs was stronger compared to free DOX. Notably, more intense fluorescence intensity was observed for DOX-loaded high Fe/Mg-CA NPs (Figure 13), indicating that the small size of both low and high Fe/Mg-CA NPs might enhance the cellular uptake through endocytosis.

On the contrary, weak fluorescence intensity was observed for free DOX, DOX-loaded CA and DOX-loaded low and high Fe/Mg-CA NPs after 24 h of treatment. This might be due to the toxicity of DOX which killed the majority of cells. Interestingly, even after 24 h, the fluorescence intensity of DOX-loaded Fe/Mg-CA NPs was more intense compared to free DOX and DOX-CA NPs (Figure 14). Thus, the small size of Fe/Mg-CA NPs facilitated the rapid internalization of DOX into the cells.

### 3.13. Quantitative Analysis of Cellular Uptake for DOX-Loaded Fe/Mg-CA NPs

The intracellular accumulation of DOX through DOX-loaded low Fe/Mg-CA and DOX-loaded high Fe/Mg-CA NPs in MCF-7 cells was further examined quantitatively by measuring the fluorescence intensity of cell lysates, through a 2030 multilabel reader victor X5. After 4 h of treatment, the fluorescence intensity of DOX-loaded low Fe/Mg-CA and DOX-loaded high Fe/Mg-CA NPs was considerably higher than that of free DOX, with the highest level of fluorescence intensity detected for DOX-loaded high Fe/Mg-CA NPs (Figure 15). The fluorescence intensity was decreased after 24 h of treatment, suggesting the cytotoxic effect of DOX, which was consistent with the earlier observation of the fluorescence micrographs. Noticeably, even after 24 h, the fluorescence intensity of DOX-loaded Fe/Mg-CA NPs was more compared to DOX-loaded CA NPs and free DOX (Figure 15). These data, again, support the notion that the small size of Fe/Mg-CA NPs facilitated better cellular uptake of the loaded drugs, leading to higher intracellular accumulation of free drugs through the rapid dissolution of the particles in acidic endosomal pH. Table 6 demonstrated the cellular uptake (%) of free DOX and DOX-loaded CA, low and high Fe/Mg-CA NPs in MCF-7 cells after 4 h and 24 h of treatment.

### 3.14. Protein Corona Analysis

Protein corona profiling by the in-solution digestion method using the Q-TOF LC-MS/MS was performed in an attempt to identify the serum proteins that have an affinity for NPs and therefore might influence the half-life, biodistribution and tumor accumulation of these particles. The serum proteins that were found to bind to CA, low Fe/Mg-CA and high Fe/Mg-CA NPs, were listed in Appendix A, pointing out their isoelectric point (pI) and major biological functions. The pie charts (Figure 16) demonstrated the taxonomy (expressed in percentage) of the proteins according to their main biological functions.

The majority of the proteins detected in the corona of CA, low Fe/Mg-CA and high Fe/Mg-CA NPs are acidic (pI ˂ 7.5) (i.e., negatively charged) in physiological pH, and therefore would electrostatically adhere to the cation-rich domains of CA (Ca^2+^) and Fe/Mg-CA (Ca^2+^, Mg^2+^, and Fe^3+^) NPs. In contrast, the proteins that were positively charged in the physiological pH would be attracted to the anionic (CO_3_^2−^ and PO_4_^3−^) regions of the CA and Fe/Mg-CA NPs.

The pie charts (Figure 16) showing the taxonomy of the proteins according to their biological functions revealed that transport proteins constituted the majority, i.e., around 60% in CA, 57% in low Fe/Mg-CA and 50% in high Fe/Mg-CA. Other proteins, such as structural proteins, cell adhesive proteins, coagulation proteins were found to interact more with Fe/Mg-CA than CA. However, overall fewer proteins were found to be associated with Fe/Mg-CA NPs than CA NPs, as reflected by the formation of smaller size pellets following centrifugation of serum-supplemented Fe/Mg-CA particles. This could be explained by the notion that the existence of Fe and Mg or the rough surface of Fe/Mg-CA NPs might promote the binding of selective proteins, such as cell adhesive proteins and reduce interactions of other proteins, such as protease inhibitor, metabolic proteins and so on.

More comprehensive tables describing the protein corona profiles in terms of molecular weight (Da), -10lgP and coverage (%) were shown in Appendix A. Serum albumin and ALB proteins (precursor of serum albumin) constituted the majority of the coverage (%) (Appendix A). Opsonins (globulin, fibrinogen and complement proteins) were not detected in the corona of CA and Fe/Mg-CA NPs. It should be noted that dysopsonins could heighten the circulation time of the NPs in vivo, while opsonins were known to promote macrophage recognition and subsequent phagocytosis of particles, eventually causing the particles to be cleared from the body by the reticuloendothelial system (RES) [14]. Having no interaction with opsonins and possessing a selective affinity for dysopsonins, cell adhesive proteins and transport proteins might play crucial roles in prolonging the blood circulation time of Fe/Mg-CA NPs, with their resultant preferential accumulation in tumor via EPR effect and efficient uptake by individual cancer cells.

### 3.15. Animal Biodistribution Study of Fe/Mg-CA NPs

Biodistribution studies help to understand the pharmacokinetics and pharmacodynamics of the drugs [59,60], and the fate of the NPs inside the body (i.e., its distribution, clearance pathway from the body and accumulation in organs) [61,62] after binding with the plasma protein. To evaluate the distribution and accumulation of DOX in tumor and different organs (e.g., brain, spleen, liver, heart, kidney, lungs, and muscle), the intravenous administration of DOX in free and NPs-bound forms was carried out in subcutaneous 4T1 tumor-induced Balb/c mice. The mice were allocated randomly into five groups (n = 3) and intravenously injected with DOX solution, DOX-CA NPs, low DOX-Fe/Mg-CA NPs and high DOX-Fe/Mg-CA NPs. Tumor and major organs were harvested at 24 h post-treatment. Figure 17 demonstrated the in vivo fluorescence intensity of different tissues in mice.

As can be seen from Figure 17, the fluorescence intensity in the tumor was significantly higher for low DOX-Fe/Mg-CA and high DOX-Fe/Mg-CA NPs compared to DOX-CA and free DOX. This was attributed to the successful accumulation of DOX-loaded CA, DOX-loaded low Fe/Mg-CA and DOX-loaded high Fe/Mg-CA in the tumor, facilitated by their successful extravasation from the leaky vasculatures through EPR effect, resulting in higher effective dose and better therapeutic efficacy. The accumulation of DOX-loaded high Fe/Mg-CA NPs was lower in most organs, indicating their low systemic toxicity. The fluorescence intensity detected for high DOX-Fe/Mg-CA NPs in the brain was significantly less compared to DOX solution, DOX-CA NPs, and low DOX-Fe/Mg-CA NPs. These results suggested that high Fe/Mg-CA NPs have less tendency to pass the blood-cerebrospinal fluid barrier (BCSFB) and blood-brain barrier (BBB) compared to CA and low Fe/Mg-CA NPs. Moreover, high DOX-Fe/Mg-CA showed no DOX deposition in the heart and kidneys, implying that the receptors present in the kidney and heart cells might not recognize the protein corona (ligands) formed with high Fe/Mg-CA NPs, hence, preventing their accumulation in these two organs. Since cardiotoxicity is the severest side effect of DOX, the distribution of DOX in the heart is undesirable. It should be noted that high Fe/Mg-CA with the particle size of 189 ± 9 nm was not subjected to renal clearance since the threshold size for glomerular filtration is typically 6–8 nm [63].

On the contrary, a markedly increased amount of DOX was detected in the heart for DOX solution, which was almost 3.6-fold greater than DOX-CA and 40.7-fold greater than low Fe/Mg-CA (Figure 17). This might be due to the reduction in the size of free DOX molecules which enable them to pass easily through the tight junctions of the cardiac muscle tissue, inducing greater cardiotoxicity compared to DOX-loaded CA and low Fe/Mg-CA NPs. Both DOX-CA and DOX-low Fe/Mg-CA have accumulated in RES organs (liver, lungs, spleen) and other tissues. However, a reduced amount of fluorescence intensity was detected for DOX-high Fe/Mg-CA NPs in the RES organs (e.g., lungs, liver). The accumulation of DOX in the liver and spleen for three different NPs might be due to their uptake by macrophages of the RES. DOX-CA, DOX-low Fe/Mg-CA and DOX-high Fe/Mg-CA NPs showed reduced accumulation in the lungs than free DOX. This might be due to the low affinity of the receptors present in the lungs towards the protein corona formed with the respective DOX-loaded NPs. Noticeably, the deposition of DOX in the spleen for DOX-high Fe/Mg-CA NPs was 6.2-fold higher than DOX-low Fe/Mg-CA NPs and 11.4-fold higher than DOX-CA NPs. This might be due to the receptors present in the surface of spleen macrophages which might have a greater affinity for the specific proteins of the corona formed with high DOX-Fe/Mg-CA NPs, resulting in its enhanced accumulation compared to DOX-CA and low DOX-Fe/Mg-CA NPs. The accumulation of DOX in non-targeted tissues, such as muscle, was significantly higher for DOX solution, DOX-loaded CA and DOX-loaded low Fe/Mg-CA NPs compared to DOX-loaded high Fe/Mg-CA NPs. DOX accumulation in the muscle might result in weakened muscular function [64], and increased muscular fatigue [65]. Therefore, less accumulation of DOX in the muscles wound prevent muscular dysfunction and toxicity.

### 3.16. Blood Plasma Analysis of Fe/Mg-CA NPs

Blood plasma analysis was conducted to determine the presence of DOX in the plasma at 24 h post-treatment with DOX solution, DOX-loaded CA, DOX-loaded low Fe/Mg-CA, and DOX-loaded high Fe/Mg-CA NPs. As shown in Figure 18, the fluorescence intensity for DOX-loaded CA, DOX-loaded low Fe/Mg-CA, and DOX-loaded high Fe/Mg-CA NPs was almost 1.6-fold, 1.8-fold, and 1.3-fold higher than that of free DOX, suggesting the enhanced circulation time of DOX in the blood for DOX-loaded CA, low Fe/Mg-CA, and high Fe/Mg-CA NPs compared to free DOX.

## 4. Conclusions

In summary, the present findings showed that both low Fe/Mg-CA and high Fe/Mg-CA NPs have a nano-sized dimension with improved homogeneity and potential ability to mitigate the self-aggregation compared to CA NPs, resulting in enhanced drug binding, more effective cellular uptake of drug-loaded NPs and higher cytotoxic effect on the cancer cells. Both of the particle types demonstrated fast dissolution and consequential rapid release of DOX in extreme endosomal acidic pH, which supported their pH-responsive nature like CA particles. Furthermore, the protein corona study indicated a higher affinity of Fe/Mg-CA NPs towards cell-adhesive proteins than CA NPs, in addition to possessing a common affinity for transport-related proteins. The blood plasma analysis confirmed that DOX-loaded CA, low Fe/Mg-CA and high Fe/Mg-CA have considerably heightened the circulation time of DOX in the blood compared to free DOX solution. These results were in line with the protein corona analysis which showed interactions of DOX-loaded CA, low Fe/Mg-CA and high Fe/Mg-CA NPs with dysopsonins, enhancing their circulation time in blood and protecting the drug cargo from degradation in blood before reaching the tumor site. The biodistribution study revealed that Fe/Mg-CA NPs could effectively deliver DOX into tumors. Compared to CA and low Fe/Mg-CA NPs, high Fe/Mg-CA NPs showed less off-target distribution in most organs. The in vivo tissue distribution of high Fe/Mg-CA NPs showed no deposition of DOX in the kidneys and cardiac heart muscles, eliminating the possibility of kidney toxicity and cardiotoxicity by DOX. Both low and high Fe/Mg-CA NPs showed less accumulation in major RES organs (liver, lungs), thus, mitigating macrophage uptake in the RES and prolonging blood circulation time. However, high accumulation of DOX was noticed in the spleen for high Fe/Mg-CA compared to low DOX-Fe/Mg-CA and DOX-CA NPs. Thus, Fe/Mg-CA NPs, especially high Fe/Mg-CA NPs, emerge as a promising anti-cancer drug delivery vehicle for treating cancers including breast cancer. However, further study in preclinical cancer models should be conducted to evaluate the huge potential of the novel Fe/Mg-CA nano-carrier system.

## Figures and Tables

**Figure 1 nanomaterials-10-00834-f001:**
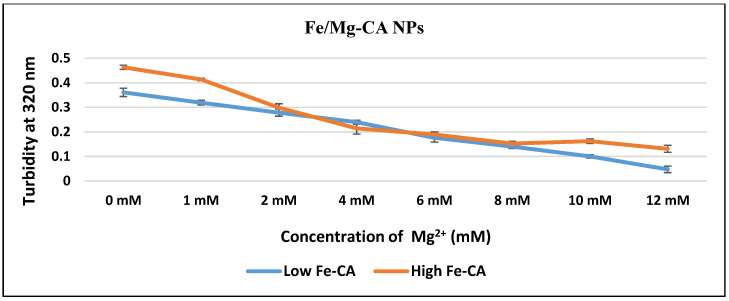
Turbidity measurement at 320 nm of Fe/Mg-carbonate apatite (Fe/Mg-CA) nanoparticles (NPs) formed by mixing increasing concentrations of exogenous Mg^2+^ (0, 2, 4, 6, 8 and 10 mM) in a fixed concentration of exogenous Ca^2+^ (4 mM Ca^2+^) and two different (low and high) concentrations of Fe^3+^, 1 µM and 5 µM in 1 mL Dulbecco’s modified eagle medium (DMEM), followed by incubation at 37 °C for 30 min and supplementation with 10% fetal bovine serum (FBS).

**Figure 2 nanomaterials-10-00834-f002:**
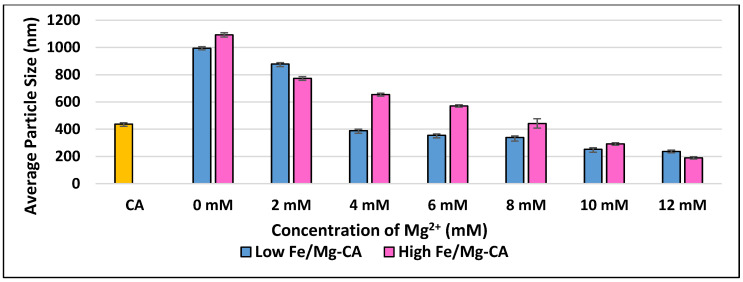
The z-average diameter of low and high Fe/Mg-CA NPs formed by mixing increasing concentrations of exogenous Mg^2+^ in a fixed concentration of exogenous Ca^2+^ (4 mM Ca^2+^) and two different (low and high) concentrations of Fe^3+^, 1 µM and 5 µM, respectively in 1 mL DMEM, prior to incubation at 37 °C for 30 min, and supplementation with 10% FBS.

**Figure 3 nanomaterials-10-00834-f003:**
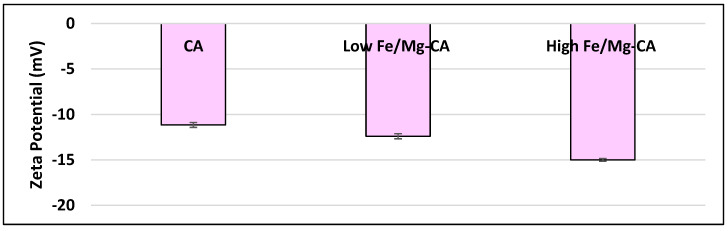
Surface charge measurement of CA, low Fe/Mg-CA and high Fe/Mg-CA NPs. The particles were generated through incubation at 37 °C for 30 min after exogenously adding to 1 mL of DMEM a fixed concentration of Ca^2+^ (4 mM Ca^2+^) for CA NPs and additionally, two different concentrations of Fe^3+^, 1 µM and 5 µM and Mg^2+^, 4 mM and 12 mM, respectively, for low Fe/Mg-CA and high Fe/Mg-CA NPs.

**Figure 4 nanomaterials-10-00834-f004:**
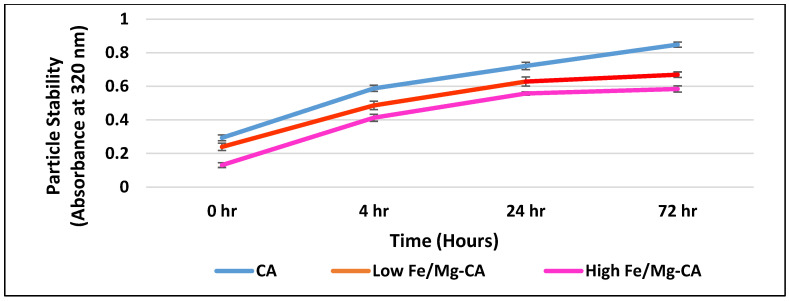
Stability analysis of CA, low Fe/Mg-CA and high Fe/Mg-CA NPs based on turbidity measurement at 320 nm over a period of 72 h. The temperature was maintained at 37 °C throughout the experiment.

**Figure 5 nanomaterials-10-00834-f005:**
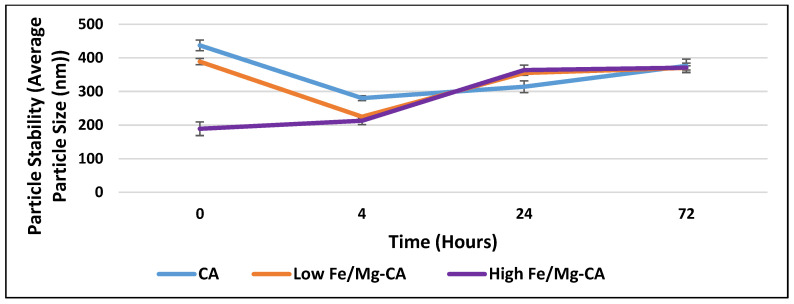
Stability analysis of CA, low Fe/Mg-CA and high Fe/Mg-CA NPs based on measurement of z-average diameter over a period of 72 h. The temperature of 37 °C was maintained throughout the experiment.

**Figure 6 nanomaterials-10-00834-f006:**
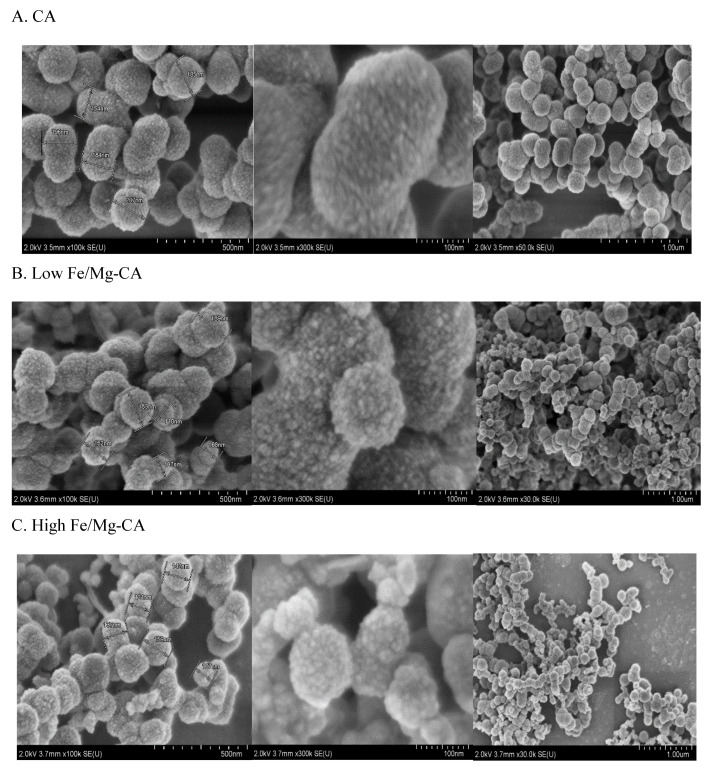
Field emission scanning electron microscopy (FESEM) micrographs in absence of 10% FBS showing size, shape and surface morphology of differently formulated particles. Scale bar: 500 nm, 100 nm, and 1 µm. (**A**) CA (**B**) Low Fe/Mg-CA (**C**) High Fe/Mg-CA. After the respective particles were incubated at 37 °C for 30 min, they were centrifuged twice for 15 min at 13,000 rpm. The supernatants were discarded and the pellets were resuspended in 20 µL Milli Q water. Then 3 µL of the respective NP suspensions was dried at room temperature for 45 min.

**Figure 7 nanomaterials-10-00834-f007:**
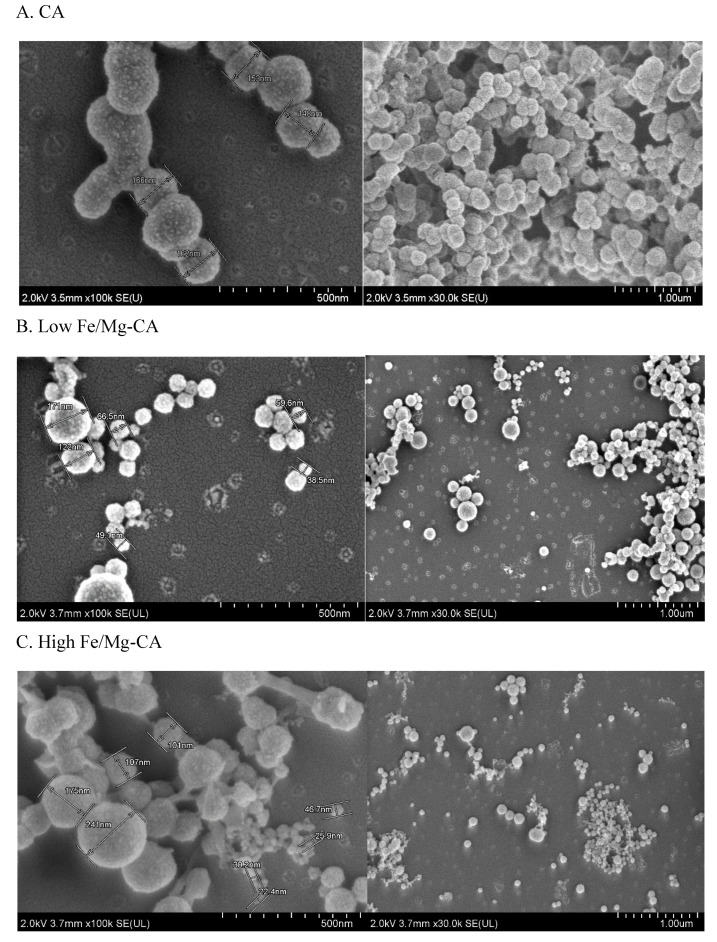
FESEM micrographs in presence of 10% FBS showing size, shape and surface morphology. Scale bar: 500 nm, and 1 µm. (**A**) CA (**B**) Low Fe/Mg-CA (**C**) High Fe/Mg-CA. After the respective particles were incubated and supplemented with 10% FBS, they were centrifuged twice for 15 min at 13,000 rpm. The supernatants were discarded and the pellets were resuspended in 20 µL Milli Q water. Then 3 µL of the respective NP suspensions was dried at room temperature for 45 min.

**Figure 8 nanomaterials-10-00834-f008:**
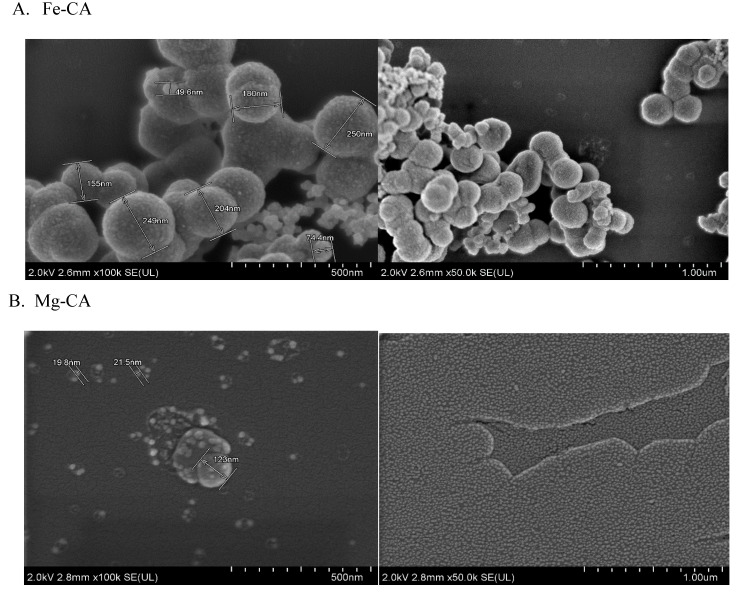
FESEM micrographs showing individual effects of Fe^3+^ and Mg^2+^ on CA NPs. Scale bar: 500 nm, and 1 µm. (**A**) Fe-CA (**B**) Mg-CA. After the respective particles were incubated and supplemented with 10% FBS, they were centrifuged twice for 15 min at 13,000 rpm. The supernatants were discarded and the pellets were resuspended in 20 µL Milli Q water. Then 3 µL of the respective NP suspensions was dried at room temperature for 45 min.

**Figure 9 nanomaterials-10-00834-f009:**
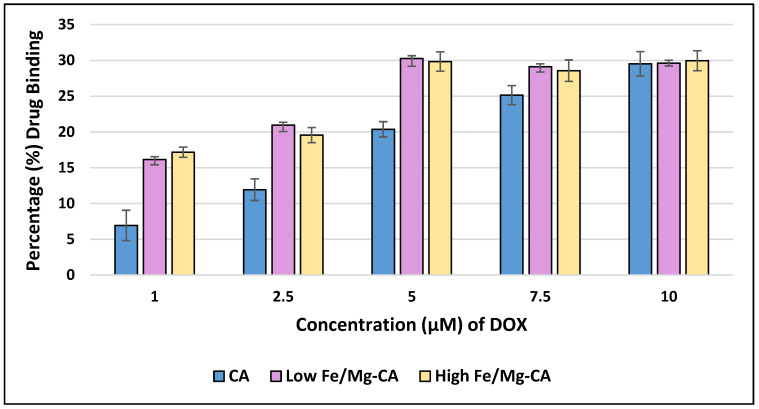
Estimation of binding affinity of doxorubicin (DOX) for CA, low Fe/Mg-CA and high Fe/Mg-CA NPs with increasing concentrations of DOX (1, 2.5, 5, 7.5 and 10 μM).

**Figure 10 nanomaterials-10-00834-f010:**
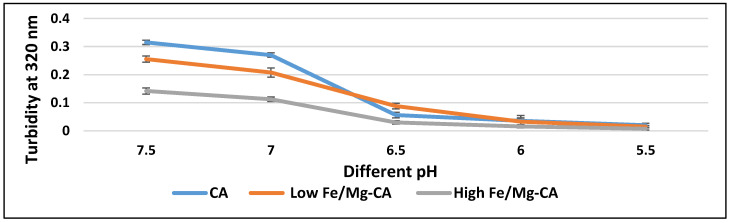
Dissolution profiles of CA (20 mM Ca^2+^), low Fe/Mg-CA (20 mM Ca^2+^, 5 µM Fe^3+^ and 20 mM Mg^2+^), and high Fe/Mg-CA (20 mM Ca^2+^, 25 µM Fe^3+^ and 60 mM Mg^2+^) NPs with decreasing pH. The particles were prepared in 200 µL freshly prepared DMEM media, prior to incubation at 37 °C for 30 min. Then 800 µL of DMEM media of different pHs (7.5, 7, 6.5, 6 and 5.5) was supplemented to each sample.

**Figure 11 nanomaterials-10-00834-f011:**
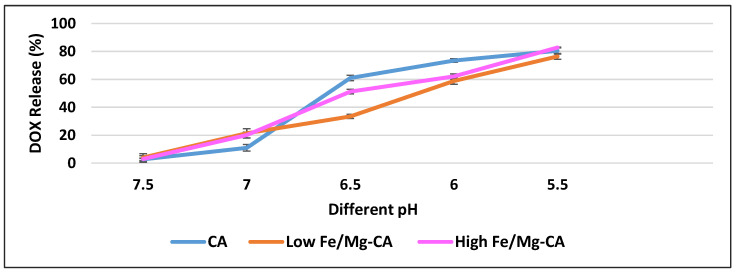
pH dependent release of DOX from CA (20 mM Ca^2+^), low Fe/Mg-CA (20 mM Ca^2+^, 5 µM Fe^3+^ and 20 mM Mg^2+^) and high Fe/Mg-CA (20 mM Ca^2+^, 25 µM Fe^3+^ and 60 mM Mg^2+^) NPs. DOX (5 µM) was used for each particle preparation in 200 µL freshly prepared DMEM media prior to incubation at 37 °C for 30 min. Then, 800 µL of DMEM media of different pHs (7.5, 7, 6.5, 6 and 5.5) was added to each sample.

**Figure 12 nanomaterials-10-00834-f012:**
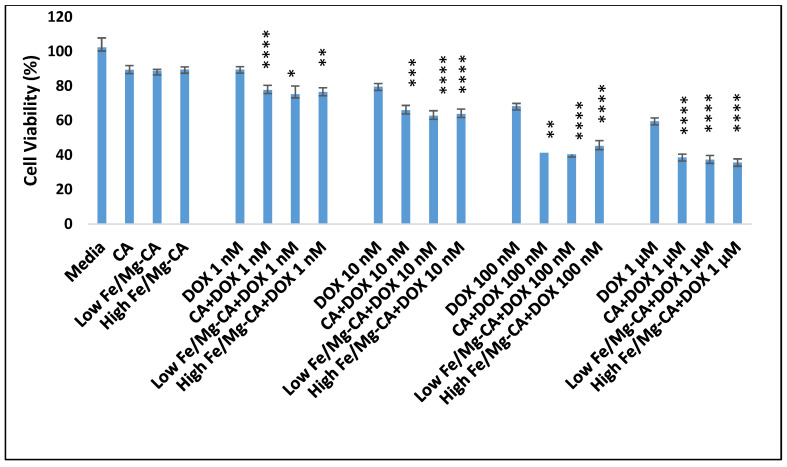
Cell viability assessment for free DOX, DOX-loaded CA, DOX-loaded low Fe/Mg-CA and DOX-loaded high Fe/Mg-CA by thiazolyl blue tetrazolium bromide (MTT) assay in Michigan Cancer Foundation-7 (MCF-7) cells after 48 h of treatment. Results were considered significant at *p*-value 0.01–0.05 (*), very significant at *p*-value 0.001–0.01 (**), highly significant at *p*-value 0.0001–0.001 (***) and extremely significant at *p*-value ˂ 0.0001 (****). *p* values were calculated with respect to control (DOX in media) and treatment (DOX-bound NPs). The confidence interval was 95% for the respective samples.

**Figure 13 nanomaterials-10-00834-f013:**
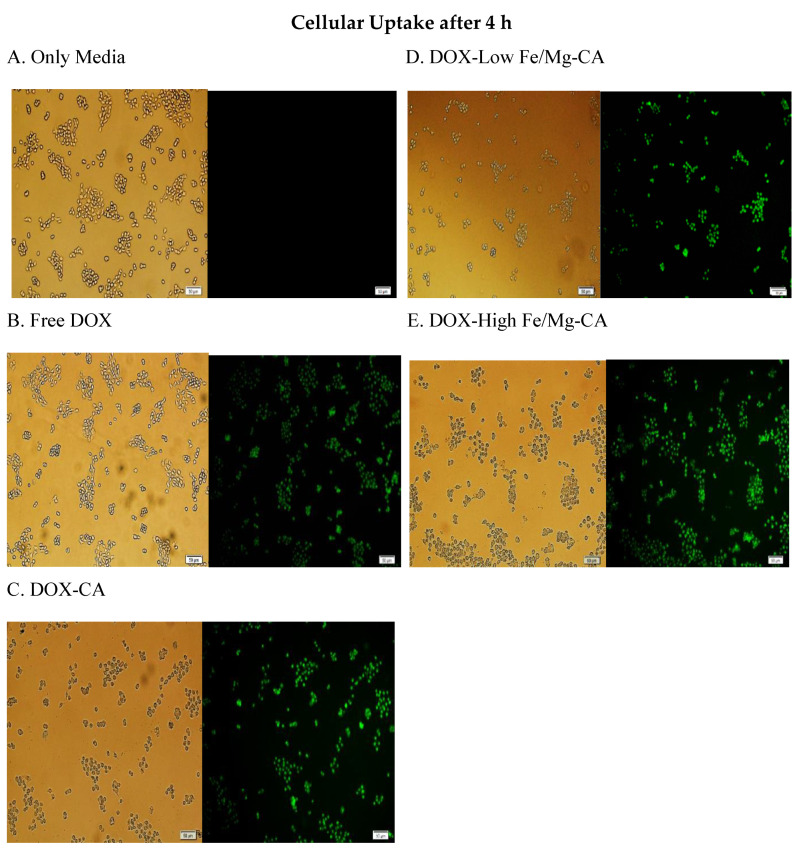
Representative images of cellular uptake in MCF-7 cells after 4 h of treatment. (**A**) Only media (**B**) Free DOX (5 µM DOX) (**C**) DOX-loaded CA (5 µM DOX, 4 mM Ca^2+^) (**D**) DOX-loaded low Fe/Mg-CA (5 µM DOX, 4 mM Ca^2+^,1 µM Fe^3+^, 4 mM Mg^2+^) (**E**) DOX-loaded high Fe/Mg-CA (5 µM DOX, 4 mM Ca^2+^, 5 µM Fe^3+^, 12 mM Mg^2+^).

**Figure 14 nanomaterials-10-00834-f014:**
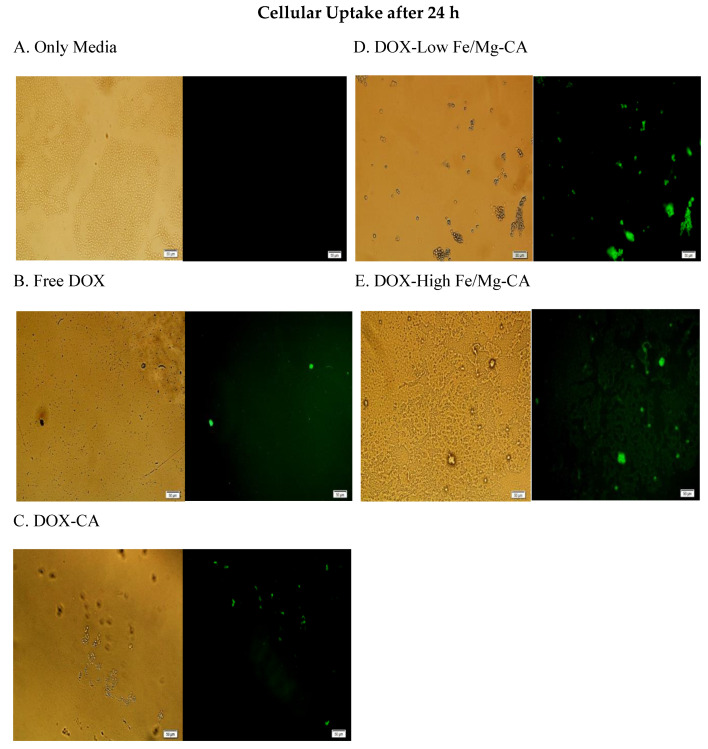
Representative images of cellular uptake in MCF-7 cells after 24 h of treatment. (**A**) Only media (**B**) Free DOX (5 µM DOX) (**C**) DOX-loaded CA (5 µM DOX, 4 mM Ca^2+^) (**D**) DOX-loaded low Fe/Mg-CA (5 µM DOX, 4 mM Ca^2+^,1 µM Fe^3+^, 4 mM Mg^2+^) (**E**) DOX-loaded high Fe/Mg-CA (5 µM DOX, 4 mM Ca^2+^, 5 µM Fe^3+^, 12 mM Mg^2+^).

**Figure 15 nanomaterials-10-00834-f015:**
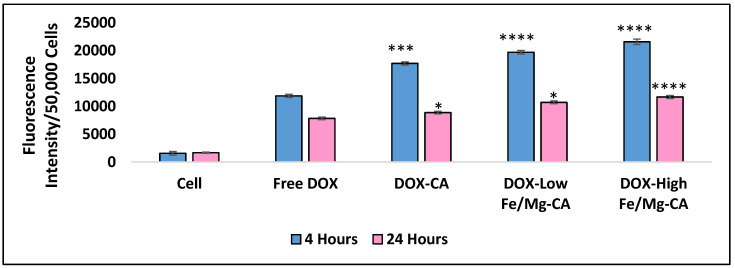
Fluorescence intensity measurement of free DOX, DOX-loaded low Fe/Mg-CA, DOX-loaded high Fe/Mg-CA and DOX-loaded CA particles in MCF-7 cells after 4 h and 24 h of treatment. Results were considered significant at *p*-value 0.01–0.05 (*), highly significant at *p*-value 0.0001–0.001 (***) and extremely significant at *p*-value ˂ 0.0001 (****). *p* values were calculated with respect to control (DOX in media) and treatment (DOX bound NPs). The confidence interval was 95% for the respective samples.

**Figure 16 nanomaterials-10-00834-f016:**
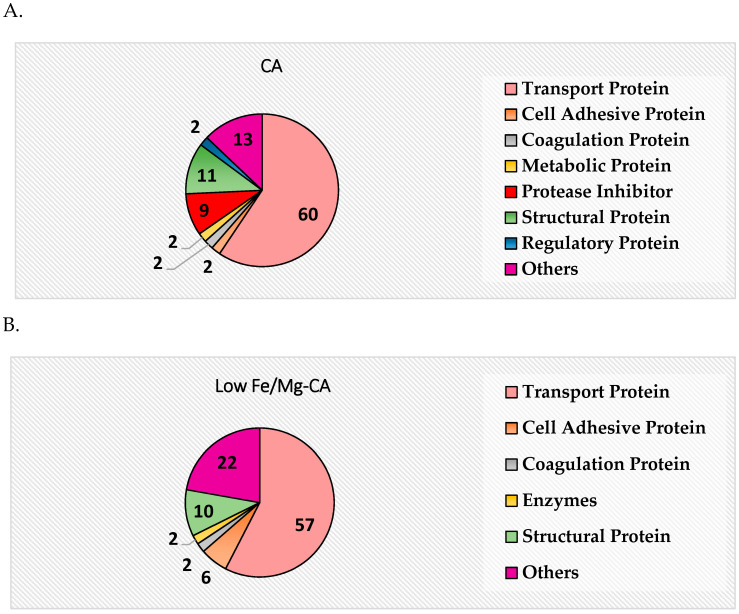
Biological function profiling of corona proteins (expressed as %) (**A**) CA (4 mM Ca^2+^). (**B**) Low Fe/Mg-CA (4 mM Ca^2+^, 1 µM Fe^3+^ and 4 mM Mg^2+^). (**C**) High Fe/Mg-CA (4 mM Ca^2+^, 5 µM Fe^3+^ and 12 mM Mg^2+^). After incubation and supplementation with 10% FBS, the particles were again incubated for 20 min at 37 °C and centrifuged twice at 13,000 rpm for 15 min. The pellets were then dissolved in 100 µL of 50 mM ethylene diamine tetraacetic acid (EDTA) in H_2_O.

**Figure 17 nanomaterials-10-00834-f017:**
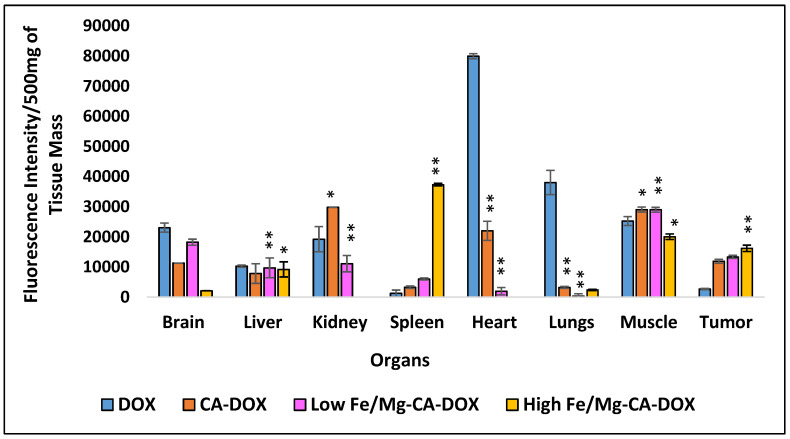
Tissue distribution of DOX in Balb/c 4T1 tumor-bearing mice after 24 h of intravenous injection with DOX solution, DOX-CA NPs, DOX-low Fe/Mg-CA NPs and DOX-high Fe/Mg-CA NPs. Results were considered significant at *p*-value 0.01–0.05 (*), very significant at *p*-value 0.001–0.01 (**). *p* values were calculated with respect to control (DOX in media) and treatment (DOX bound NPs). The confidence interval was 95% for the respective samples.

**Figure 18 nanomaterials-10-00834-f018:**
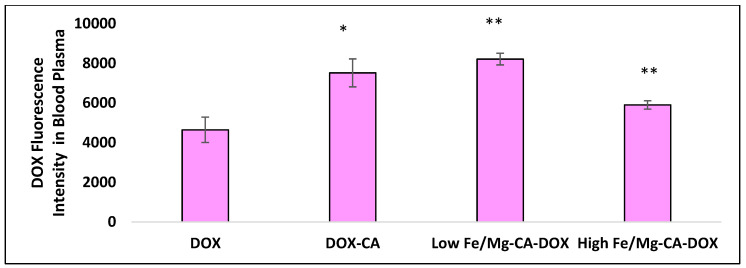
DOX fluorescence intensity in the blood plasma of Balb/c 4T1 tumor-bearing mice after 24 h of intravenous injection with DOX solution, DOX-CA NPs, low DOX-Fe/Mg-CA NPs and high DOX-Fe/Mg-CA NPs for DOX solution. Results were considered significant at *p*-value 0.01–0.05 (*), very significant at *p*-value 0.001–0.01 (**). *p* values were calculated with respect to control (DOX in media) and treatment (DOX bound NPs). The confidence interval was 95% for the respective samples.

**Table 1 nanomaterials-10-00834-t001:** Required parameters for the detection of Ca^2+^, Fe^3+^ and Mg^2+^.

Element	Wavelength (nM)	Flame
Ca	422.7	Air- Acetylene
Fe	248.3	Air- Acetylene
Mg	285.2	Air- Acetylene

**Table 2 nanomaterials-10-00834-t002:** Fourier transform infrared spectroscopy (FTIR) profile for CA (carbonate apatite), and low Fe/Mg-CA (low Fe/Mg-carbonate apatite) and high Fe/Mg-CA NPs (high Fe/Mg-carbonate apatite).

Sample	Wave Number (cm^−1^)	Ionic Groups
CA	865, 1435, 1474, 1662	CO_3_^2−^
	668, 1012	PO_4_^3−^
High Fe/Mg-CA	862, 1436, 1488, 1520, 1652	CO_3_^2−^
	566, 668, 1006	PO_4_^3−^
Low Fe/Mg-CA	866, 1436, 1488, 1508, 1662	CO_3_^2−^
	555, 575, 668, 1017	PO_4_^3−^

**Table 3 nanomaterials-10-00834-t003:** Estimation of Ca/P and Ca+Mg/P ratios (wt.(%) ± standard deviation) of low and high Fe/Mg-CA NPs.

Ratio	Low Fe/Mg-CA (wt. (%) ± stdev.)	High Fe/Mg-CA (wt. (%) ± stdev.)	CA (wt. (%) ± stdev.)
Ca/P	3.65 ± 0.11	4.27 ± 0.22	2.00 ± 0.13
Ca+Mg/P	5.08 ± 0.13	5.81 ± 0.22	2.39 ± 0.09

**Table 4 nanomaterials-10-00834-t004:** Comparison of the amount of Mg^2+^ substitutes for Ca^2+^ in low and high Fe/Mg-CA NPs.

Sample	Mg/Ca (wt.(%) ± stdev.)
Low Fe/Mg-CA	0.41 ± 0.06
High Fe/Mg-CA	0.41 ± 0.01
CA	0.16 ± 0.01

**Table 5 nanomaterials-10-00834-t005:** Weight (%) of Ca^2+^ in CA, and Ca^2+^, Fe^3+^ and Mg^2+^ in both low and high Fe/Mg-CA NPs.

Sample	Weight (%) of Ca ± stdev.	Weight (%) of Mg ± stdev.	Weight (%) of Fe ± stdev.
CA	29 ± 0.07		
Low Fe/Mg-CA	20 ± 0.16	1 ± 0.05	0.3 ± 0.08
High Fe/Mg-CA	25 ± 0.14	3 ± 0.11	2 ± 0.26

**Table 6 nanomaterials-10-00834-t006:** Cellular uptake (%) of free DOX, DOX-loaded CA, DOX-loaded low and high Fe/Mg-CA NPs in MCF-7 cells after 4 h and 24 h of treatment.

Samples	Cellular Uptake (%) (4 h after Treatment)	Cellular Uptake (%) (24 h after Treatment)
Free DOX	26 ± 0.28	17 ± 0.01
DOX-loaded low Fe/Mg-CA	44 ± 0.42	24 ± 0.97
DOX-loaded high Fe/Mg-CA	49 ± 0.42	26 ± 0.61
DOX-loaded CA	40 ± 1.14	19 ± 0.67

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
