# Peer review of "Fe/Mg-Modified Carbonate Apatite with Uniform Particle Size and Unique Transport Protein-Related Protein Corona Efficiently Delivers Doxorubicin into Breast Cancer Cells"

_nanomaterials, 2020, doi:10.3390/nano10050834_

Round 1
Reviewer 1 Report
The authors have taken the reviewers suggestion from previous review.
I think sufficient changes have been made.
The manuscript can be accepted in present form.
Author Response
Thank you very much.
Reviewer 2 Report
The authors evidently revised the manuscript and added a very interesting set of in vivo data, showing the location of the DOX-loaded particles in different organs. Even then, however, the discussion contains some errors. For example, the authors comment that “the accumulation of DOX in the liver, lungs and spleen for three different NPs might be due to their
uptake by macrophages of the RES”, but a close look at organ distribution data in Fig.18 shows much lesser and minimal accumulation in lungs for DOX-NP samples than that claimed by the authors.
Overall, this manuscript contains very good sets of data, but they are presented in a way that is very scattered and not focused. I agree with the editorial remark that some figures that are virtually impossible to use to see anything meaningful (such as optical images of nanoparticles) and that they should be moved to the supplementary section, but that has not been done. Also, the abstract may be now too long and the figure captions are very poor and do not do a good job providing an accurate and correct description of what the figures show. Discussion, in addition, as I noticed in the first review, is solid whenever it relates to biological phenomena, but very poor when it relates to materials science phenomena and Nanomaterials is, first and foremost, a materials journal.
Author Response
- The authors evidently revised the manuscript and added a very interesting set of in vivo data, showing the location of the DOX-loaded particles in different organs. Even then, however, the discussion contains some errors. For example, the authors comment that “the accumulation of DOX in the liver, lungs and spleen for three different NPs might be due to their uptake by macrophages of the RES”, but a close look at organ distribution data in Fig.18 shows much lesser and minimal accumulation in lungs for DOX-NP samples than that claimed by the authors.
Answer: The explanation with regards to less accumulation of DOX loaded NPs in the lungs has been revised (Section 3.15)
- Overall, this manuscript contains very good sets of data, but they are presented in a way that is very scattered and not focused. I agree with the editorial remark that some figures that are virtually impossible to use to see anything meaningful (such as optical images of nanoparticles) and that they should be moved to the supplementary section, but that has not been done.
Answer: Optical images have been moved to the Supplementary section (Supplementary Figure 3).
- Also, the abstract may be now too long and the figure captions are very poor and do not do a good job providing an accurate and correct description of what the figures show.
Answer: The abstract has been shortened. Also, description has been provided in figure captions (Figure 1 to 11, 13, 14, and 16).
Plagarism:
We have revised the writing. The edited write-up has been highlighted in red within the text.
We would like to thank the reviewer for his valuable comments and suggestions to improve the manuscript. Please let us know if you have any more comments or suggestions.
Round 2
Reviewer 2 Report
none
Author Response
Thank you very much.
We have checked and corrected the grammatical mistakes.